# Targeting sex determination to suppress mosquito populations

Ming Li[1], Nikolay P Kandul[1], Ruichen Sun[1], Ting Yang[1], Elena D Benetta[1], Daniel J Brogan[1], Igor Antoshechkin[2], Héctor M Sánchez C[3], Yinpeng Zhan[4], Nicolas A DeBeaubien[4], YuMin M Loh[5], Matthew P Su[5,6], Craig Montell[4], John M Marshall[3,7], Omar S Akbari[1]*

[1]School of Biological Sciences, Department of Cell and Developmental Biology, University of California, Berkeley, Berkeley, United States; [2]Division of Biology and Biological Engineering (BBE), California Institute of Technology, Pasadena, United States; [3]Divisions of Epidemiology & Biostatistics, School of Public Health, University of California, Berkeley, Berkeley, United States; [4]Department of Molecular, Cellular, and Developmental Biology and the Neuroscience Research, Institute, University of California, Santa Barbara, Santa Barbara, United States; [5]Graduate School of Science, Nagoya University, Nagoya, Japan; [6]Institute for Advanced Research, Nagoya University, Nagoya, Japan; [7]Innovative Genomics Institute, Berkeley, United States

*For correspondence:
oakbari@ucsd.edu

**Abstract** Each year, hundreds of millions of people are infected with arboviruses such as dengue, yellow fever, chikungunya, and Zika, which are all primarily spread by the notorious mosquito *Aedes aegypti*. Traditional control measures have proven insufficient, necessitating innovations. In response, here we generate a next-generation CRISPR-based precision-guided sterile insect technique (pgSIT) for *Ae. aegypti* that disrupts genes essential for sex determination and fertility, producing predominantly sterile males that can be deployed at any life stage. Using mathematical models and empirical testing, we demonstrate that released pgSIT males can effectively compete with, suppress, and eliminate caged mosquito populations. This versatile species-specific platform has the potential for field deployment to effectively control wild populations of disease vectors.

## eLife assessment

This **valuable** paper builds on a method, previously conceptualized and validated, of genetic control for insect populations. The method, called pgSIT, uses integrated CRISPR-Cas9 based constructs to generate, in certain combinations of genotypes, mutations that cause both male sterility and female inviability. Release of such genotypes in sufficiently large numbers can lead to an inundation of a local insect population with sterile males and this can lead to localised population suppression, which represents an effective method of control for problematic insect populations. The data are **convincing** and will be of interest to anyone working on vector control strategies.

## Introduction

The mosquito *Aedes aegypti* is the principal vector of several deadly viruses, including dengue, yellow fever, chikungunya, and Zika. Unfortunately, its geographic spread has been exacerbated by climate change, global trade, and its preference for anthropogenic habitats, such that 3.9 billion people are presently at risk (*Brady et al., 2012*). Traditionally, mosquito control has been achieved through insecticides, though developing resistance has rendered them less effective. Moreover, insecticides are harmful to non-target species, such as insect pollinators, and direct exposure can be dangerous for

humans and pets (*Hemingway et al., 2002*; *Lu et al., 2020*; *Sánchez-Bayo, 2021*; *Trdan, 2016*). Alternative integrated approaches for mosquito control include preventative measures designed to reduce mosquito populations through habitat reduction or larval predation, though these can also be impractical or introduce invasive species. Notwithstanding, the continuous expansion of this invasive mosquito species demonstrates that current measures are insufficient.

Alternative genetic biological approaches have been developed to control specific species, including the Sterile Insect Technique (SIT), the *Wolbachia*-mediated Incompatible Insect Technique (WIIT), Female-Specific Release of Insects Carrying a Dominant Lethal gene (fsRIDL), Inherited Female Elimination by Genetically Encoded Nucleases to Interrupt Alleles (IFEGENIA), Gene Drive (GD) and precision-guided SIT (pgSIT) (*Knipling, 1955*; *Laven, 1967*; *Spinner et al., 2022*; *Smidler et al., 2023*; *Li et al., 2017*; *Kandul et al., 2019b*; *Li et al., 2021*). Here, radiation-sterilized males (♂'s), or ♂'s harboring *Wolbachia*, are repeatedly mass released to mate with wild females (♀'s) which typically mate only once, thereby reducing mosquito populations over time. The fsRIDL can genetically sex-sort ♂'s, and can be released as eggs. IFEGENIA disrupts a female-essential gene, femaleless (fle), demonstrating complete genetic sexing via heritable daughter gynecide. GD leverage the CRISPR based precise DNA cutting and repair to spread into populations. Two types of gene drives have been engineered: suppression drives collapses a population, whereas modification drives spread cargo genes that can potentially block the chain of transmission of a targeted disease (*Champer et al., 2016*; *Marshall et al., 2019*; *Raban and Akbari, 2017*; *Wang et al., 2021*). The pgSIT works by simultaneously disrupting genes essential for ♀ viability and ♂ fertility during development to genetically generate sex-sorted and viable sterilized ♂'s.

Previously, we showed that disrupting a ♀-specific flight muscle gene (fem-myo) induced a flightless ♀-specific phenotype in *Ae. aegypti* ♀'s (*Li et al., 2021*). However, this approach would still result in viable flightless transgenic ♀'s being released which may be less desirable than removing females altogether. To further build upon the system, we instead cause ♀ lethality, simplifying the process of generating pgSIT ♂'s for release. Our herein-developed next-generation pgSIT approach simultaneously disrupts three genes in *Ae. aegypti*: *doublessex* (*dsx*), *intersex* (*ix*), and *β-Tubulin 85D* (*βTub*), to induce ♀ lethality, or intersex (⚥) transformation, and near-complete ♂/♀ sterility. We demonstrate that pgSIT ♂'s are competitive and multiple releases can suppress cage populations. Notably, mathematical modeling indicates that this technology can control insect populations effectively over a wide range of realistic performance parameters and fitness costs. Overall, this next-generation pgSIT approach provides a viable option for the release of sterile males, bringing the field significantly closer to deployable CRISPR-based vector control strategies. Additionally, we also develop a rapid diagnostic assay to detect pgSIT transgene presence that could be used alongside field tests. Taken together, we have demonstrated that pgSIT could be designed to target essential sex determination and fertility genes in *Ae. aegypti* to produce sterilized males that can suppress and eliminate caged populations thereby providing an alternate system for controlling this deadly vector.

## Results

### Disrupting sex determination and fertility

To induce ♀-specific lethality and/or ⚥ transformation in *Ae. aegypti*, we targeted either the *dsx* (AAEL009114) or *ix* (AAEL010217) genes, both of which are associated with sex determination and differentiation in insects (*Chase and Baker, 1995*; *Garrett-Engele et al., 2002*; *Kyrou et al., 2018*; *Mysore et al., 2015*; *Salvemini et al., 2011*; *Scali et al., 2005*; *Trujillo-Rodríguez et al., 2021*). The *dsx* gene is located on chromosome 1R and is alternatively spliced to produce two ♀-specific (*dsxF₁* and *dsxF₂*) and one ♂-specific (*dsxM*) isoforms (*Salvemini et al., 2011*). Exon 5b is retained in both ♀-specific *dsxF₁* and *dsxF₂* transcripts and is spliced out in the ♂-specific *dsxM* transcript. Therefore, to disrupt both ♀-specific *dsx* isoforms, we selected four gRNAs that target exon 5b (*Supplementary file 1a*) and assembled the pBac *gRNA^dsx* plasmid (*Figure 1A*). The *ix* gene (AAEL010217) is located on chromosome 3L and is orthologous to the *ix* gene required for ♀-specific sexual differentiation in *Drosophila melanogaster* (*Chase and Baker, 1995*). While the *D. melanogaster ix* does not undergo sex-specific alternative splicing and is expressed in both sexes, *ix* interacts with *dsxF*, but not *dsxM*, resulting in ♀-specific activity (*Garrett-Engele et al., 2002*). To disrupt *ix*, we engineered a *gRNA^ix* cassette harboring two gRNAs targeting *ix* exon 1 (*Figure 1A*, *Supplementary file 1a*). To determine

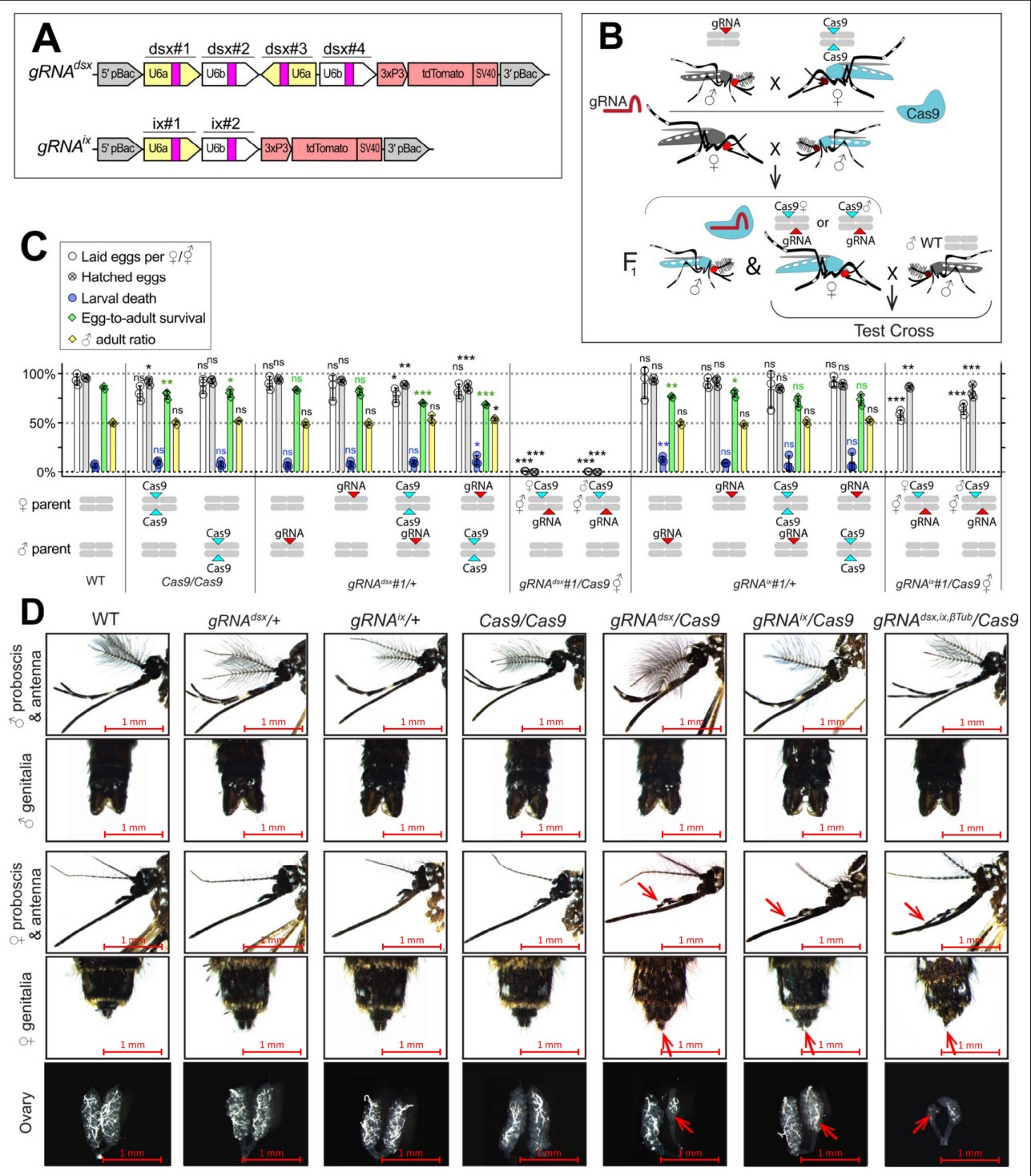

**Figure 1.** CRISPR/Cas9-mediated disruption of *dsx* or *ix* affects female mosquito morphology and fecundity. (**A**) Schematic map of gRNA genetic constructs. The *gRNA^dsx^* and *gRNA^ix^* constructs harbor a *3xP3-tdTomato* marker and multiple gRNAs guiding Cas9 to the female-specific *doublesex* (*dsx*) gene or the female-active *intersex* (*ix*) gene, respectively. (**B**) A schematic of the genetic cross between the homozygous *Cas9* and hemizygous *gRNA^dsx^/+* or *gRNA^ix^/+* mosquitoes to generate trans-hemizygous $F_1$ females (♀'s). Reciprocal genetic crosses were established and two types of trans-hemizygous $F_1$ ♀'s were generated: *gRNA/+; ♀Cas9/+* ♀'s inherited a maternal Cas9; and *gRNA/+; ♂Cas9/+* ♀'s inherited a paternal Cas9. Then, both trans-hemizygous ♀'s were crossed to wild-type (WT) males (♂'s), and their fecundity was assessed. A comparison of the fecundity and male ratio of trans-hemizygous, hemizygous *Cas9* or *gRNA* ♀'s to those of WT ♀'s. The bar plot shows means and one standard deviation (± SD) over triple biological replicates (*n* = 3). The data presentsone transgenic strain of each construct, *gRNA^dsx^#1* and *gRNA^ix^#1*, as all strains induced similar results (all data can be

*Figure 1 continued on next page*

*Figure 1 continued*

found in *Supplementary file 1c*). (**D**) All *gRNA^dsx^#1/+; Cas9/+* and *gRNA^ix^#1/+; Cas9/+* trans-hemizygous ♀ mosquitoes exhibited male-specific features (red arrows in **D**), had reduced fecundity, and were transformed into intersexes (♀'s). Statistical significance of mean differences was estimated using a two-sided Student's *t*-test with equal variance (ns: $p \geq 0.05$, *$p < 0.05$, **$p < 0.01$, and ***$p < 0.001$). Source data are provided in *Supplementary file 1*.

The online version of this article includes the following figure supplement(s) for figure 1:

**Figure supplement 1.** Changes in morphological structures induced by disruption of targeted genes.

**Figure supplement 2.** Induced mutations at the targeted sequences in *dsx* and *ix*.

**Figure supplement 3.** *Ae. aegypti dsx* or *ix* disruption does not result in significant alterations in ear anatomy, wing beat frequency (WBF), and male phonotactic behaviors.

the effects of targeting these genes, multiple separate *gRNA^dsx^* and *gRNA^ix^* strains were generated and assessed by crossing with *Nup50-Cas9* (hereafter, *Cas9*) (*Li et al., 2021*). To explore potential effects of maternal Cas9 carryover, all genetic crosses were conducted reciprocally (*Figure 1B*). Remarkably, the disruption of either *dsx* or *ix* transformed all trans-hemizygous ♀'s into intersexes (♀'s), independent of whether Cas9 was inherited from the mother (i.e., ♀Cas9), or father (i.e., ♂Cas9). These *gRNA^dsx^/+; Cas9/+* ♀ exhibited multiple malformed morphological features, such as mutant maxillary palps, abnormal genitalia, and malformed ovaries (*Figure 1D*, *Figure 1—figure supplement 1*, *Supplementary file 1b*). Notably, only ~50% of *gRNA^dsx^/+; Cas9/+* ♀'s were able to blood feed, producing very few unhatchable eggs. While a similar fraction of *gRNA^ix^/+; Cas9/+* ♀ were able to blood feed, the fed ♀'s laid hatchable eggs (*Figure 1C*, *Supplementary file 1c*). Sanger sequencing confirmed expected mutations at the *dsx* and *ix* loci (*Figure 1—figure supplement 2*). Importantly, we did not observe changes in the ear anatomy, wing beat frequency (WBF), or phonotactic behaviors in either *gRNA^dsx^/+; Cas9/+* ♂'s or *gRNA^ix^/+; Cas9/+* ♂'s suggesting that the trans-hemizygous ♂'s have normal courtship behavior (*Figure 1—figure supplement 3*, *Supplementary file 1c*). Taken together, these findings reveal that ♀ transformation can be efficiently achieved by disrupting either *dsx* or *ix* genes, while the ♀ or ♀ sterility requires the *dsx* disruption.

## Multiplexed target gene disruption

To guide CRISPR-mediated disruption of multiple genes essential for ♀-specific viability and ♂-specific fertility, we engineered the *gRNA^dsx,ix,βTub^* construct which combines the effective gRNAs described above (*Figure 1—figure supplement 2*). More specifically, this construct harbors six gRNAs: three gRNAs targeting the *dsx* exon 5b; one gRNA targeting the *ix* exon 1; and two gRNA targeting the *β-Tubulin 85D* (*βTub*, AAEL019894) exon 1 (*Li et al., 2021*; *Figure 2A*, *Supplementary file 1a*). The *βTub* gene is specifically expressed in mosquito testes (*Akbari et al., 2013*; *Degner et al., 2019*; *Gamez et al., 2020*) and is essential for sperm maturation, elongation, and motility (*Chen et al., 2021*; *Li et al., 2021*). Three independent homozygous *gRNA^dsx,ix,βTub^* strains were established and reciprocally crossed to the homozygous *Cas9* strain to measure survival, fitness, and fertility of generated pgSIT (i.e., trans-hemizygous) progeny (*Figure 2B, C*, *Figure 2—figure supplement 1*). To determine transgene integration sites, we performed genome sequencing of *gRNA^dsx,ix,βTub^#1* (hereafter, *gRNA^dsx,ix,βTub^*) and found that this strain harbored a single copy of the transgene inserted on chromosome 3 (*Figure 2—figure supplement 2*). Using these sequencing data, we also validated the presence of diverse mutations at the intended target sites (*Figure 2—figure supplement 3*). RNA sequencing also revealed the presence of mutations in the target RNA sequences (*Figure 2—figure supplements 4–6*, *Supplementary file 1j, k*). These findings reveal that single copies of *Cas9* and *gRNA^dsx,ix,βTub^* can induce mutagenesis at six separate DNA targets simultaneously, thereby providing desired pgSIT phenotypic redundancy for increased resilience.

## pgSIT ♀'s and ♀'s are incapacitated

We next tested the simultaneous disruption of *dsx*, *ix*, and *βTub*, by reciprocally crossing *gRNA^dsx,ix,βTub^* with *Cas9*, which resulted in the death of most pgSIT ♀'s and ♀'s, as no ♀'s and only 5–6% ♀'s survived to adulthood (*Figure 2C*). Interestingly, around 95% of pgSIT ♀'s failed to eclose, independent of whether Cas9 was inherited from the mother or father (termed *pgSIT^♀Cas9^* or *pgSIT^♂Cas9^* ♀'s, respectively, *Supplementary file 1d*). The pgSIT ♀'s that did eclose had more pronounced abnormal phenotypes than ♀'s generated from *dsx* or *ix* single-gene disruption (*Figure 1D*, *Figure 1—figure*

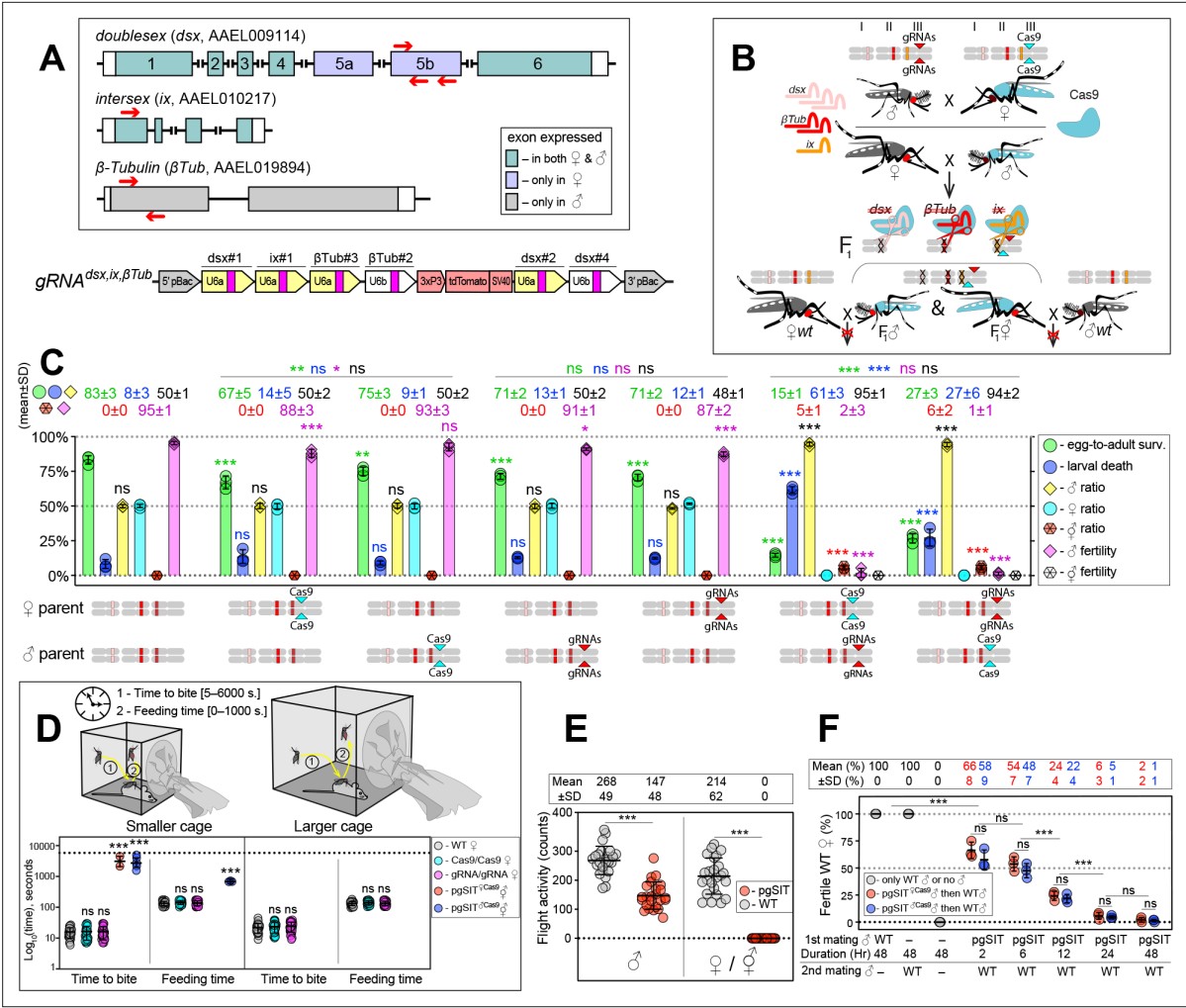

**Figure 2.** The precision-guided sterile insect technique (pgSIT) cross results in nearly complete female lethality and male sterility. (**A**) Schematic maps of targeted genes (box) and *gRNA*$^{dsx,ix,\beta Tub}$ construct. Red arrows show relative locations of gRNA target sequences (box). *gRNA*$^{dsx,ix,\beta Tub}$ harbors a *3xP3-tdTomato* marker and six gRNAs to guide the simultaneous CRISPR/Cas9-mediated disruption of *dsx*, *ix*, and *βTub* genes. Violet exon Box 5a and 5b represent ♀-specific exons. (**B**) A schematic of the reciprocal genetic cross between the homozygous *Cas9*, marked with *Opie2-CFP*, and homozygous *gRNA*$^{dsx,ix,\beta Tub}$ to generate the trans-hemizygous F$_1$ (aka. pgSIT) progeny. Relative positions of *dsx*, *ix*, and *βTub* target genes (bar color corresponds to gRNA color), and transgene insertions in the *Cas9* (*Nup50-Cas9* strain[1]) and *gRNA*$^{dsx,ix,\beta Tub}$*#1* strains are indicated in the three pairs of *Ae. aegypti* chromosomes. To assess the fecundity of generated pgSIT mosquitoes, both trans-hemizygous ♀'s and ♂'s were crossed to the wild-type (WT) ♂'s and ♀'s, respectively. (**C**) Comparison of the survival, sex ratio, and fertility of trans-hemizygous, hemizygous *Cas9* or *gRNA*$^{dsx,ix,\beta Tub}$*#1*, and WT mosquitoes. The bar plot shows means ± standard deviation (SD) (*n* = 3, all data in **Supplementary file 1d**). (**D**) Blood feeding assays using both types of trans-hemizygous intersexes (♀'s): *pgSIT*$^{♀Cas9}$ and *pgSIT*$^{♂Cas9}$ ♀'s. To assess blood feeding efficiency, individual mated ♀'s or ♀'s were allowed to blood feed on an anesthetized mouse inside a smaller (24.5 × 24.5 × 24.5 cm) or larger cage (60 × 60 × 60 cm), and we recorded the time: (1) to initiate blood feeding (i.e., time to bite); and (2) of blood feeding (i.e., feeding time). The plot shows duration means ± SD over 30 ♀'s or ♀'s (*n* = 30) for each genetic background (**Supplementary file 1e**). (**E**) Flight activity of individual mosquitoes was assessed for 24 hr using vertical *Drosophila* activity monitoring (DAM, **Supplementary file 1f**). The plot shows means ± SD (n=24). (**F**) Mating assays for fertility of offspring produced via crosses between trans-hemizygous ♂'s that inherited a maternal *Cas9* (*pgSIT*$^{♀Cas9}$) or paternal *Cas9* (*pgSIT*$^{♂Cas9}$) and WT ♀'s (**Supplementary file 1h**). The plot shows fertility means ± SD over three biologically independent groups of 50 WT ♀'s (*n* = 3) for each experimental condition. Statistical significance of mean differences was estimated using a two-sided Student's *t*-test with equal variance (ns: p ≥ 0.05, *p < 0.05, **p < 0.01, and ***p < 0.001). Source data are provided in **Supplementary file 1**.

The online version of this article includes the following figure supplement(s) for figure 2:

**Figure supplement 1.** Genetic characterization of independent transgenic strains for CRISPR/Cas9-mediated disruption of *dsx*, *ix*, and *βTub*.

**Figure supplement 2.** Determination of transgene copy number for the *gRNA*$^{dsx,ix,\beta Tub}$*#1* strain using Oxford Nanopore genome sequencing.

**Figure supplement 3.** Oxford Nanopore sequencing results validating disruptions in the target sites.

**Figure supplement 4.** Integrative genome browser snapshot of *dsx* validating gRNA target disruption in both the DNA and RNA.

*Figure 2 continued on next page*

**Figure supplement 5.** Integrative genome browser snapshot of *ix* validating gRNA target disruption in both the DNA and RNA.

**Figure supplement 6.** Integrative genome browser snapshot of *βTub* validating gRNA target disruption in both the DNA and RNA.

**Figure supplement 7.** Transcription profiling and expression analysis of *Ae. aegypti* liverpool and precision-guided sterile insect technique (pgSIT) mosquito samples.

**Figure supplement 8.** Longevity, fecundity, and developmental times of the generated precision-guided sterile insect technique (pgSIT) mosquitoes.

---

*supplement 1*) and were sterile (*Figure 2C*, *Supplementary file 1d*). Moreover, the vast majority of adult pgSIT ♀'s had reduced fitness as they could not blood feed (*Figure 2D*, *Supplementary file 1e*), did not fly as frequently (*Figure 2E*, *Supplementary file 1f*), had smaller abdomens and wings (*Figure 1—figure supplement 1*, *Supplementary file 1b*), and had 4x-reduced longevity (*Figure 2— figure supplement 8A, B*, *Supplementary file 1g*). Taken together, these results demonstrate that *Cas9* in combination with *gRNA*$^{dsx,ix,βTub}$ induces either the lethality or transformation of pgSIT ♀'s into sterile unfit ♀'s.

## pgSIT ♂'s are mostly sterile with normal longevity and mating ability

We found that ~94% of the eclosed pgSIT mosquitoes were ♂'s due to the majority of the ♀'s not surviving to adulthood (*Figure 2C*, *Supplementary file 1d*). Both *pgSIT*$^{♀cas9}$ and *pgSIT*$^{♂Cas9}$ ♂'s were also nearly completely sterile (*Figure 2C*, *Supplementary file 1d*). Surprisingly, despite our assumption that *pgSIT*$^{♀cas9}$♂ and *pgSIT*$^{♂Cas9}$ ♂ would be roughly equivalent, the *pgSIT*$^{♀cas9}$ ♂'s showed less flight activity than wild-type (WT) (*Figure 2E*, *Supplementary file 1f*), and *pgSIT*$^{♀cas9}$ ♂'s developmental times were slightly delayed (*Figure 2—figure supplement 8D*). Nevertheless, the longevity of pgSIT ♂'s was not significantly different from WT ♂'s (*Figure 2—figure supplement 8*, *Supplementary file 1g*), and both *pgSIT*$^{♀cas9}$♂ and *pgSIT*$^{♂Cas9}$ ♂ were competitive and mated with WT ♀'s to effectively suppress the fertility and re-mating ability of WT ♀'s (*Figure 2F*, *Supplementary file 1h*). Taken together, these findings reveal that maternal deposition of Cas9 may induce some fitness costs, however these could be avoided by using a paternal source of Cas9, and despite these observed fitness effects, both *pgSIT*$^{♀cas9}$♂ and *pgSIT*$^{♂Cas9}$ ♂ were competitive and both types could likely be capable of inducing population suppression.

## pgSIT ♂'s induce population suppression

To further explore whether pgSIT ♂'s are competitive enough to mate with WT ♀'s in the presence of WT ♂'s to suppress populations, we conducted multigenerational, population cage experiments. At each generation, adult *pgSIT*$^{♀cas9}$ or *pgSIT*$^{♂Cas9}$ and WT ♂ mosquitos were released into cages using several introduction frequencies (pgSIT:WT – 1:1, 5:1, 10:1, 20:1, and 40:1; *Figure 3A* and *Supplementary file 1i*). Remarkably, both *pgSIT*$^{♀cas9}$ ♂ or *pgSIT*$^{♂Cas9}$ ♂ behaved similarly, and the high-release ratios (20:1, 40:1) eliminated all populations in five generations (*Figure 3B, C*). With lower release ratios of 10:1 and 5:1, we also observed significant suppression of the rates of laid and hatched eggs in the cages. No effect was observed in control cages harboring WT mosquitoes with only Cas9 or gRNA mosquitoes (*Figure 3B, C* and *Supplementary file 1i*). To confirm the sterility of pgSIT ♂'s, hatched larvae were screened for the presence of transgenesis markers each generation. In only 3 out of 30 experimental cages, we scored some larvae harboring transgene markers, though this was very few, indicating that the majority of *pgSIT* ♂ were indeed sterile (*Supplementary file 1i*). These findings confirmed the ability of both generated *pgSIT*$^{♀cas9}$ and *pgSIT*$^{♂Cas9}$ ♂'s to compete with WT ♂'s for mating with WT ♀'s to gradually suppress and eliminate targeted caged populations.

## Theoretical performance of pgSIT in a wild population

To explore the effectiveness of non-optimal pgSIT gene disruption on population suppression outcomes, we simulated releases of pgSIT eggs into a population of 10,000 *Ae. aegypti* adults (*Figure 4*). Weekly releases of up to 500 pgSIT eggs (♀ and ♂) per WT adult (♀ and ♂) were simulated over 1–52 weeks. The scale of releases was chosen considering adult release ratios of 10:1 are common for sterile ♂ mosquito interventions (*Carvalho et al., 2015*) and ♀ *Ae. aegypti* produce >30 eggs per day in temperate climates (*Mordecai et al., 2019*). We assume released eggs have the same survival probability as wild-laid eggs; however if released eggs do have higher mortality, this

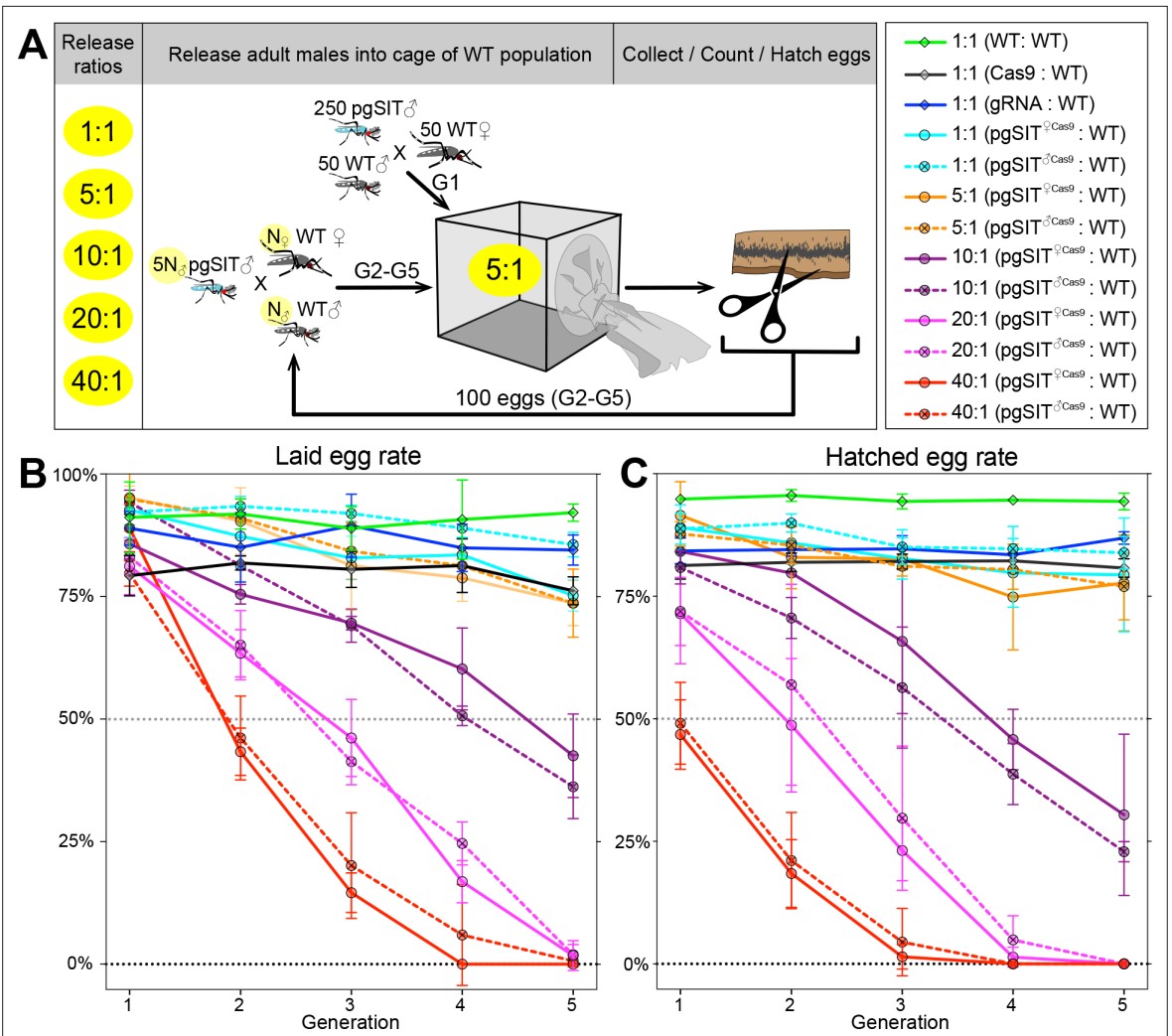

**Figure 3.** Population suppression in multigenerational population cage experiments. (**A**) Multiple precision-guided sterile insect technique (pgSIT):wild-type (WT) release ratios, such as 1:1, 5:1, 10:1, 20:1, and 40:1, were tested in triplicate. A schematic diagram depicts the cage experiments for the 5:1 ratio. To start the first generation (G1) of multigenerational population cages, 250 mature pgSIT adult ♂'s were released with 50 similarly aged WT adult ♂'s into a cage, and in 1 hr 50 virgin ♀'s were added. The mosquitoes were allowed to mate for 2 days before all ♂'s were removed, and ♀'s were blood fed and eggs were collected. All eggs were counted, and 100 eggs were selected randomly to seed the next generation (G2). The sex-sorted adult ♂'s that emerged from 100 eggs ($N_\sigma$) were released with five times more ($5N_\sigma$) pgSIT adult ♂'s into a cage, and $N_\varphi$ ♀'s were added later. The remaining eggs were hatched to measure hatching rates and score transgene markers. This procedure was repeated for five generations for each cage population (**Supplementary file 1i**). Two types of pgSIT ♂'s were assessed: those that inherited a maternal *Cas9* (*pgSIT^♀Cas9*), and those that inherited a paternal *Cas9* (*pgSIT^♂Cas9*). Multigeneration population cage data for each release ratio plotting the normalized percentage of laid eggs (**B**) and hatched eggs (**C**). Points show means ± standard deviation (SD) for each triplicate sample (n=3). Source data are provided in **Supplementary file 1**.

would be equivalent to considering a smaller release. A preliminary sensitivity analysis was conducted to determine which parameters population suppression outcomes are most sensitive to (**Figure 4—figure supplement 1**). The analysis considered probability of elimination (the percentage of 60 stochastic simulations that result in *Ae. aegypti* elimination) and window of protection (the time duration for which ≥50% of 60 stochastic simulations result in ≥90% suppression of the *Ae. aegypti* population) as outcomes, and explored how these vary with number of releases, size of releases, release interval, gRNA cutting rate, Cas9 maternal deposition rate, pgSIT ♂ fertility and mating competitiveness, and pgSIT ♀ viability. Suppression outcomes were found to be most sensitive to release schedule parameters (number, size, and interval of releases), ♂ fertility, and ♀ viability. Model-predicted efficacy of pgSIT-induced population suppression was then explored in more detail as a function of these parameters through a series of simulations depicted in **Figure 4**. Results from these simulations reveal

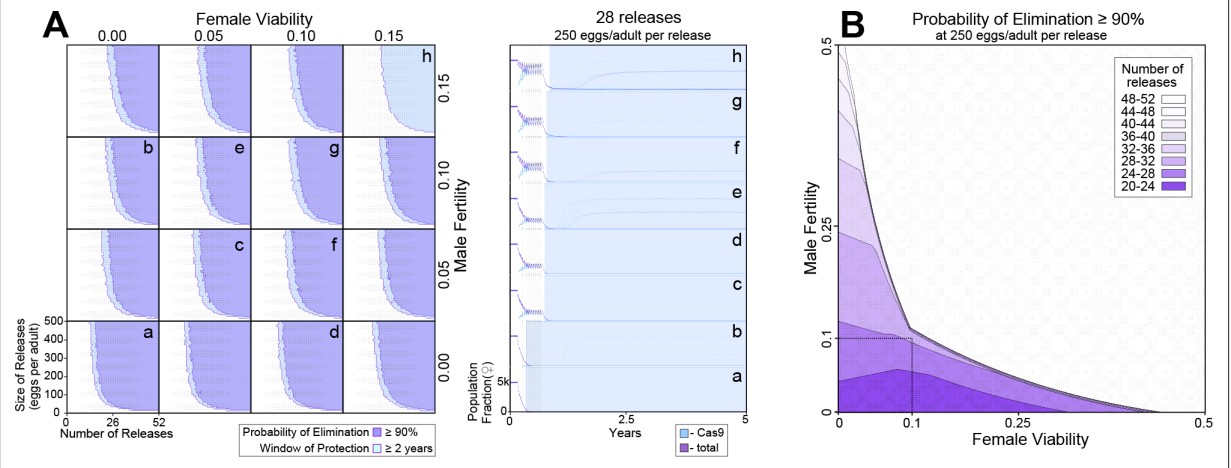

**Figure 4.** Model-predicted efficacy of precision-guided sterile insect technique (pgSIT) egg releases on *Ae. aegypti* population suppression as a function of release scheme, male sterility, and female viability. Weekly releases were simulated in a randomly mixing population of 10,000 adult mosquitoes using the MGDrivE simulation framework[2] and parameters described in ***Supplementary file 1l***. Population suppression outcomes were identified as being most sensitive to model parameters describing the release scheme, male fertility, and female viability (***Figure 4—figure supplement 1***). (**A**) These parameters were varied in factorial experiments assessing suppression outcomes including probability of elimination (the percentage of 60 stochastic simulations that result in *Ae. aegypti* elimination), and window of protection (the time duration for which ≥50% of 60 stochastic simulations result in ≥90% suppression of the *Ae. aegypti* population). Female viability was varied between 0 (complete inviability) and 0.15, male fertility was varied between 0 (complete sterility) and 0.15, release size was varied between 0 and 500 eggs released per wild adult, and the number of weekly releases was varied between 0 and 52. Regions of parameter space for which the probability of elimination exceeds 90% are depicted in purple, and in which the window of protection exceeds two years in light blue. Time-series of population dynamics for select parameter sets are depicted in a–h. Here, the total female population is denoted in red, and the Cas9-carrying female population is denoted in blue. The light blue shaded region represents the window of protection. Imperfect female inviability and male sterility result in lower probabilities of elimination; however the window of protection lasts for several years for male fertility and female viability in the range 0–0.15 for simulated release schemes. (**B**) Regions of parameter space for which the probability of elimination exceeds 90% are depicted as a function of male fertility (*x*-axis), female viability (*y*-axis), and the minimum number of weekly releases required to achieve this (shadings, see key). Release size is set to 250 eggs per wild adult. The shaded square depicts the region of parameter space in which male fertility is between 0% and 10% and female viability is between 0% and 10%. A ≥90% elimination probability is achieved with ~20–32 weekly releases for pgSIT systems having these parameters. Source data are provided in ***Supplementary file 1***.

The online version of this article includes the following figure supplement(s) for figure 4:

**Figure supplement 1.** Sensitivity of precision-guided sterile insect technique (pgSIT) population suppression outcomes to model parameters.

**Figure supplement 2.** Sensitive and rapid detection of transgenic DNA fragments with the sensitive enzymatic nucleic acid sequence reporter (SENSR) assay.

that a window of protection exceeding 2 years (a common benchmark for interruption of disease transmission) is generally achieved for pgSIT ♀ viability between 0 (complete inviability) and 0.15, and pgSIT ♂ fertility between 0 (complete sterility) and 0.15, provided the release scheme exceeds ~23 releases of ≥100 pgSIT eggs per wild adult (***Figure 4A***). Achieving a ≥90% probability of elimination places slightly tighter restrictions on ♀ viability and ♂ fertility – a safe ballpark being ♀ viability and ♂ fertility both in the range 0–0.10, given a release scheme of ~26 releases of 250 pgSIT eggs per wild adult (***Figure 4B***). These results suggest a target product profile for pgSIT to be ♀ viability and ♂ fertility both in the range 0–0.10. The pgSIT strain described here falls well within these bounds, with ♀ viability of 0 and ♂ fertility of ~0.01.

## Mitigation tools for rapid transgene detection

Being able to rapidly identify transgenes in the environment in mixtures of captured mosquitoes will be important for field applications of genetic biocontrol tools. While PCR-based tests do exist, these may be difficult to implement in the field. Therefore, developing an assay with a simple readout would be desired. To generate a rapid assay for transgene detection that could be used alongside future pgSIT field trials, we adapted the sensitive enzymatic nucleic acid sequence reporter (SENSR) (***Brogan et al., 2021***) to rapidly detect pgSIT DNA fragments in *Cas9* and *gRNA*[dsx,ix,βTub] constructs in mosqui-toes. In SENSR, the presence of transgenic sequences triggers the activation of the *Ruminococcus*

*flavefaciens* CasRx/gRNA collateral activity that cleaves a quenched probe, resulting in a rapid fluorescent readout (**Brogan et al., 2021**; **Dalla Benetta et al., 2023**; **Brogan et al., 2021**). Multiple ratios of trans-hemizygous *pgSIT* to WT mosquitoes – 0:1; 1:1; 1:5; 1:10; 1:25; 1:50, and a no template control (NTC) were used to challenge SENSR. We found that SENSR effectively detected transgenic sequences from both *Cas9* and *gRNA*$^{dsx,ix,\beta Tub}$ constructs even at the lowest dilution, and no signal was observed in the WT-only and NTC controls (**Figure 4—figure supplement 2**). Notably, in every tested ratio, SENSR rapidly detected the transgenic sequence, reaching the half-maximum fluorescence (HMF) in less than 20min (**Figure 4—figure supplement 2**). Taken together, these results validated SENSR as a rapid and sensitive assay for detection of transgenic sequences in pools of mosquitoes which may be an instrumental tool for future field trials.

## Discussion

To develop a robust and stable population control technology for addressing insect disease vectors, we engineered a next-generation pgSIT system that induced near-complete ♀-specific lethality and ♂ sterility by simultaneously targeting the *dsx*, *ix*, and *βTub* genes. Our cage experiments indicate that releases of pgSIT ♂'s can induce robust population suppression and elimination, and mathematical models reinforce this finding. To support future field trials of this and other trangene-based technologies, we also developed a rapid and sensitive method, SENSR, for identifying pgSIT transgenes.

Overall, the released pgSIT males were competitive in our cage populations, as repeated releases resulted in population elimination at several different introduction frequencies. That said, we did observe some limitations. First, we determined that pgSIT males were not 100% sterile, with an estimated ~1% still producing some progeny, this may stem from using fewer gRNAs to target *βTub*, as our previous system that saw complete sterility used four gRNAs while here we used only two (**Li et al., 2021**). That said, it would not be difficult to build new strains that harbor additional gRNAs targeting *βTub* or other genes important for male fertility to ensure sterility is achieved. Even though mutations could potentially become fixed within the natural population, the likelihood of off-target effects becoming fixed within the population is exceedingly low. To mitigate potential negative impacts, we employed CHOPCHOP V3.0.0 (https://chopchop.cbu.uib.no) for the selection of gRNAs, specifically to minimize the occurrence of genomic off-target cleavage events. Furthermore, our releasing process will be carried out in multiple rounds. Even in the event that an undesired mutant is introduced into the local population, it will either be completely eradicated through subsequent rounds of releases or be naturally eliminated through the process of natural selection over time. Second, we also noticed some reduced fitness parameters when Cas9 was inherited maternally, which may result from an overabundance of Cas9 present in the egg (**Kandul et al., 2020**). That said, these maternal fitness effects can be avoided by ensuring Cas9 is inherited paternally as pgSIT phenotypes can be achieved when Cas9 is inherited from either the father or mother. Moreover, the longevity was not significantly altered, and released pgSIT males still performed well in the population cages, which indicates that these fitness costs and lack of complete male sterility could be tolerated for population suppression. Even in the presence of pgSIT construct imperfections, our mathematical modeling showed that population suppression is very promising, both in terms of the potential to eliminate a mosquito population, or to suppress it to an extent that would largely interrupt disease transmission. Our models determined that having ♀ viability and ♂ fertility both in the range 0–10% provides a reasonable target product profile for pgSIT – criteria that the construct described in this study easily satisfies. Previously published analyses (**Li et al., 2021**) have also shown fitness costs associated with pgSIT constructs are well tolerated in the range 0–25%. Overall, our results indicate that perfection is not required for an effective pgSIT system, though improvements to the presented and other systems could allow for reduced introduction ratios.

Before a possible field deployment, there are still several considerations that must be addressed for this pgSIT system (**Kandul et al., 2019b**). First, our herein-presented system requires a genetic cross to produce the releasable insects, similar to previous pgSIT systems (**Kandul et al., 2022**; **Kandul et al., 2019a**; **Li et al., 2021**) that will require a scalable genetic sexing system. While sophisticated mechanical systems that exploit sex-specific morphological differences have been generated in previous work (**Crawford et al., 2020**) and are available at https://www.senecio-robotics.com/, these approaches may be cost prohibitive. Therefore, there exists a need to develop inexpensive and scalable genetic sexing systems that could be combined with pgSIT to eliminate its reliance on mechanical adult sexing

approaches. Alternatively, if a temperature inducible Cas9 strain was generated in *Ae. aegypti* then it may be possible to generate a temperature inducible pgSIT (TI-pgSIT) system as was done previously (*Kandul et al., 2021*) and this would alleviate the need for an integrated genetic sexing approach. Second, studies are required to determine the survival and mating competitiveness of released pgSIT males under field conditions, and to optimize their release protocol. Third, although pgSIT is self-limiting and inherently safe, it does require genetic modification, so regulatory authorizations will be necessary prior to implementation. Despite this, we anticipate that obtaining such authorizations will not be insurmountable.

Altogether, we demonstrate that removing females by disrupting sex determine genes is possible with pgSIT, which can inform the development of future such systems in related species. The self-limiting nature of pgSIT offers a controllable alternative to technologies such as gene drives that will persist and uncontrollably spread in the environment. Moving forward, pgSIT could offer an efficient, scalable, and environmentally friendly next-generation technology for controlling wild mosquito populations, leading to widespread prevention of human disease transmission.

## Methods

### Mosquito rearing and maintenance

*Ae. aegypti* mosquitoes used in all experiments were derived from the *Ae. aegypti* Liverpool strain, which was the source strain for the reference genome sequence (*Matthews et al., 2018*). Mosquito rearing and maintenance were performed following previously established procedures (*Li et al., 2021*).

### Guide RNA design and assessment

To induce ♀-specific lethality, *doublesex* (*dsx*, AAEL009114) and *Intersex* (*ix*, AAEL010217) genes were targeted for CRISPR/Cas9-mediated disruption. For each target gene, the DNA sequences were initially identified using reference genome assembly, and then genomic target sites were validated using PCR amplification and Sanger sequencing (see *Supplementary file 1a*). CHOPCHOP V3.0.0 (https://chopchop.cbu.uib.no) was used to select the gRNA target sites. In total, we selected four *dsx* gRNAs and two *ix* gRNAs, and assembled the *gRNA$^{dsx}$* and *gRNA$^{ix}$* constructs harboring the corresponding gRNAs. To confirm the activity of the gRNAs in vivo, we generated the transgenic strains harboring either the *gRNA$^{dsx}$* or *gRNA$^{ix}$* constructs, and crossed each *gRNA* strain to the *Cas9* strain (*Li et al., 2021*). We sequenced DNA regions targeted by each selected gRNA in F$_1$ trans-hemizygous (aka. *gRNA/+; Cas9/+*) mosquitoes to assess gRNA/Cas9-induced mutagenesis.

### Genetic construct design and plasmid assembly

All plasmids were cloned using Gibson enzymatic assembly (*Gibson et al., 2009*). DNA fragments were either amplified from available plasmids or synthesized gBlocks (Integrated DNA Technologies) using Q5 Hotstart Start High-Fidelity 2X Master Mix (New England Biolabs, Cat. #M0494S). The generated plasmids were transformed into Zymo JM109 chemically competent *E. coli* (Zymo Research, Cat. #T3005), amplified, isolated (Zymo Research, Zyppy plasmid miniprep kit, Cat. #D4036), and Sanger sequenced. Final selected plasmids were maxi-prepped (Zymo Research, ZymoPURE II Plasmid Maxiprep kit, Cat. #D4202) and sequenced thoroughly using Oxford Nanopore Sequencing at Primordium Labs (https://www.primordiumlabs.com). All plasmids and annotated DNA sequence maps are available at https://www.addgene.org/ under accession numbers: 200252 (*gRNA$^{dsx}$*, 1067B) 200251 (*gRNA$^{ix}$*, 1055J), 200253 (*gRNA$^{dsx,ix,\beta Tub}$*, 1067L), and 164846 (*Nup50-Cas9* or *Cas9*, 874PA).

### Engineering transgenic strains

Transgenic strains were generated by microinjecting preblastoderm stage embryos (0.5–1 hr old) with a mixture of the *piggyBac* plasmid (100 ng/μl) and a transposase helper plasmid (*pHsp-piggyBac*, 100 ng/μl). Embryonic collection, microinjections, transgenic lines generation, and rearing were performed following previously established procedures (*Bui et al., 2020*; *Li et al., 2021*; *Li et al., 2020*).

### Genetic assessment of gRNA strains

To evaluate the activity of *gRNA$^{dsx}$* and *gRNA$^{ix}$* strains, we reciprocally crossed hemizygous mosquitoes of each generated strains to the homozygous *Nup-Cas9* strain (*Li et al., 2021*), and thoroughly

examined $F_1$ trans-hemizygous ♀ progeny (*Supplementary file 1b, c*). Each of three established $gRNA^{dsx,ix,\beta Tub}$ strains was subjected to single-pair sibling matings over five to seven generations to generate the homozygous stocks. Zygosity was confirmed genetically by repeated test crosses to WT. To generate trans-hemizygous progeny with a maternal *Cas9* ($pgSIT^{\female Cas9}$) and paternal *Cas9* ($pgSIT^{\male Cas9}$), homozygous $gRNA^{dsx,ix,\beta Tub}$ ♂'s and ♀'s were mated to homozygous *Cas9* ♀'s and ♂'s, respectively. Then, the generated trans-hemizygous mosquitoes were mated to WT mosquitoes of the opposite sex, and numbers of laid and hatched eggs were scored to measure the fecundity. Multiple genetic control crosses were also performed for comparisons: WT ♂ × WT ♀; WT ♂ × *Cas9* ♀; *Cas9* ♂ × WT ♀; gRNA ♂ × *Cas9* ♀; gRNA ♀ × *Cas9* ♂; gRNA ♀ × WT ♂, and gRNA ♂ × WT ♀. For the genetic cross, adult mosquitoes were allowed to mate in a cage for 4–5 days, then blood meals were provided, and eggs were collected and hatched. To calculate the normalized percentage of laid eggs per ♀ (i.e., fecundity), the number of laid eggs per ♀ for each genotype replicate was divided by the maximum number of laid eggs per ♀ (*Supplementary file 1d*). The percentage of egg hatching (i.e., fertility) was estimated by dividing the number of laid eggs by the number of hatched eggs. Egg-to-adult survival ratios were the number of laid eggs divided by the total number of eclosed adults. Larvae-to-adult survival rates were calculated by dividing the number of eclosed adults by the number of larvae. Pupae–adult survival rates were calculated by dividing the number of pupae by the number of eclosed adults. Blood feeding rates were calculated by dividing the number of blood-fed ♀'s or ♂'s by the total number of ♀ or ♂'s. Trans-hemizygous ♀'s had many morphological features visibly different from those of WT ♀'s. To assess differences in external and internal anatomical structures, we dissected twenty trans-hemizygous and control mosquitoes for each genotype and sex (♂, ♀, and ⚥) in 1% phosphate-buffered saline (PBS) buffer and imaged structures on the Leica M165FC fluorescent stereomicroscope equipped with a Leica DMC4500 color camera. The measurements were performed using Leica Application Suite X (LAS X by Leica Microsystems) on the acquired images (*Supplementary file 1b*).

## Validation of induced mutations

To confirm that the ♀ lethality or transformation correlated with specific mutations at the target loci, *dsx* and *ix* gRNA targets were PCR amplified from the genomic DNA extracted from trans-hemizygous and control mosquitoes. PCR amplicons were purified directly or gel purified using Zymoclean Gel DNA Recovery Kit (Zymo Research Cat. #D4007) and then Sanger sequenced. Presence of induced mutations was inferred as precipitous drops in sequence read quality by analyzing base peak chromatograms (*Figure 1—figure supplement 2*). The primers used for PCR and sequencing, including gRNA target sequences, are listed in *Supplementary file 1a*. In addition, induced mutations and their precise localization to six target sites in *ix*, *dsx*, and *βTub* genes were confirmed by sequencing genomes of $gRNA^{dsx,ix,\beta Tub}$#1/*Cas9* ♂'s and ⚥'s using Oxford Nanopore (see below) and aligning them to the AaegL5 genome (GCF_002204515.2). Mutations in the DNA using Nanopore data and mutations in the RNA using RNAseq data were visualized using an integrated genome browser (*Figure 2—figure supplements 4–6*).

## Ear Johnston's organ microanatomy and immunohistochemistry

The heads of 2-day-old mosquitoes were removed, fixed in 4% paraformaldehyde (PFA) and 0.25% Triton X-100 PBS, and sectioned using a Leica VT1200S vibratome. The anti-SYNORF1 3C11 primary monoclonal antibody (AB_528479, 1:30, Developmental Studies Hybridoma Bank [DSHB], University of Iowa) and two secondary antibodies, Alexa Fluor 488-conjugated anti-mouse IgG (#A-11029, 1:300, Thermo Fisher) and anti-horseradish peroxidase 555 (anti-HRP, AB_2338959, 1:100, Jackson Immuno Research), were used for immunohistochemistry. All imaging was conducted using a laser-scanning confocal microscope (FV3000, Olympus). Images were taken with a silicone-oil immersion ×60 Plan-Apochromat objective lens (UPlanSApo, NA = 1.3) as previously described (*Su et al., 2018*).

## WBF measurements

Groups of 30 mosquitoes were transferred to 15 cm³ cages containing a microphone array (Knowles NR-23158-000) and provided with a source of 10% glucose water. After 2 days of LD entrainment, audio recordings of mosquito flight behavior were made using a Picoscope recording device (Picoscope 2408B, Pico Technology) at a sampling rate of 50 kHz. These recordings lasted 30 min during

periods of peak activity during dusk (ZT12.5–13). Audio recordings were then processed and analyzed using the simbaR package in R (*Somers et al., 2022*; *Su et al., 2020*). Four separate cages of mosquitoes were tested for each genotype. Statistical comparisons were made using analysis of variance (ANOVA) on Ranks tests, followed by pairwise Wilcoxon tests with a Bonferroni correction applied for multiple comparisons.

## Phonotaxis assays

Individual ♂'s were provided with a 475-Hz pure tone stimulus calibrated to 80 dB SPL ($V \approx 5 \times 10^{-4}$ ms$^{-1}$) provided for 30 s by a speaker. Prior to tone playback, a gentle puff of air was provided to each ♂ to stimulate flight initiation. Mosquitoes were considered to demonstrate phonotaxis behaviors if they orientated to the speaker and then demonstrated abdominal bending behaviors. A video camera (GoPro) was used to record all responses to stimulation, with responses being scored as either 0 (no approach/no bending) or 1 (approach and bending). Mosquitoes were tested on two consecutive days, with mosquitoes showing behavior considered as a phonotactic response to stimuli on at least one test day being regarded as responders. Three repeats were conducted for each genotype (across 3 generations), with at least 10 ♂'s tested per repeat. Statistical comparisons were made using Chi-squared tests.

## Transgene copy number and genomic integration site

To infer the transgene copy number(s) and insertion site(s), we performed Oxford Nanopore sequencing of the trans-hemizygous *gRNA$^{dsx,ix\beta Tub}$#1/Cas9* mosquitoes. Genomic DNA was extracted from 10 ♂'s and ♀'s using Blood & Cell Culture DNA Midi Kit (QIAGEN, Cat. #13343) and the sequencing library was prepared using the Oxford Nanopore SQK-LSK110 genomic DNA by ligation kit. The library was sequenced for 72 hr on MinION flowcell (R9.4.1) followed by base calling with Guppy v6.3.2 and produced 16.18 Gb of filtered data with Q score cutoff of 10. To determine transgene copy number(s), reads were mapped to the AaegL5 genome supplemented with transgene sequences *gRNA$^{dsx,ix,\beta Tub}$* (1067L plasmid) and *Cas9* (874PA plasmid) using minimap2 and sequencing depth was calculated using samtools. To identify transgene integration sites, nanopore reads were mapped to the *gRNA$^{dsx,ix,\beta Tub}$* and *Cas9* plasmids and the sequences that aligned to the expected regions between the *piggyBac* sites were parsed. These validated reads were then mapped to the AaegL5 genome (GCF_002204515.2) and sites of insertions were identified by examining the alignments in the Interactive Genomics Viewer (IGV) browser. The *gRNA$^{dsx,ix,\beta Tub}$#1* strain harbors a single copy of *gRNA$^{dsx,ix,\beta Tub}$* inserted on chromosome 3 (+/+orientation) between 2993 and 16524 at NC_035109.1:362,233,930 TTAA site (*Figure 2—figure supplement 2*). We also confirmed the previously identified insertion site of *Cas9* (*Li et al., 2021*): chromosome 3 (+/+orientation) between 2933 and 14573 at NC_035109.1:33,210,107 TTAA.

## Transcriptional profiling and expression analysis

To quantify target gene reduction and expression from transgenes as well as to assess global expression patterns, we performed Illumina RNA sequencing. We extracted total RNA using miRNeasy Mini Kit (QIAGEN, Cat. #217004) from 20 sexed pupae: WT ♀'s, WT ♂'s, trans-hemizygous *pgSIT$^{♀Cas9}$* ♂'s, and *pgSIT$^{♀Cas9}$* ♀'s in biological triplicate (12 samples total), following the manufacturer's protocol. DNase treatment was conducted using DNase I, RNase-free (Thermo Fisher Scientific, Cat. #EN0521), following total RNA extraction. RNA integrity was assessed using the RNA 6000 Pico Kit for Bioanalyzer (Agilent Technologies #5067-1513), and mRNA was isolated from ~1 µg of total RNA using NEB Next Poly (A) mRNA Magnetic Isolation Module (NEB #E7490). RNA-seq libraries were constructed using the NEBNext Ultra II RNA Library Prep Kit for Illumina (NEB #E7770) following the manufacturer's protocols. Libraries were quantified using a Qubit dsDNA HS Kit (Thermo Fisher Scientific #Q32854), and the size distribution was confirmed using a High Sensitivity DNA Kit for Bioanalyzer (Agilent Technologies #5067-4626). Libraries were sequenced on an Illumina NextSeq2000 in paired end mode with the read length of 50 nt and sequencing depth of 20 million reads per library. Base calling and FASTQ conversion were performed with DRAGEN 3.8.4. The reads were mapped to the AaegL5.0 (GCF_002204515.2) genome supplemented with the *Cas9* and *gRNA$^{dsx,ix,\beta Tub}$* plasmid sequences using STAR aligner. On average, ~97.8% of the reads were mapped. Gene expression was then quantified using featureCounts against the annotation release 101 GTF downloaded from NCBI (GCF_

002204515.2_AaegL5.0_genomic.gtf). TPM values were calculated from counts produced by feature-Counts. PCA and hierarchical clustering were performed in R and plotted using the ggplot2 package. Differential expression analyses between pgSIT vs WT samples within each sex were performed with DESeq2 (*Figure 4—figure supplement 2* and *Supplementary file 1j, k*). Gene Ontology overrepresentation analyses were done with the topGO package.

## Blood feeding assays

Ten mature WT ♀'s or pgSIT ♀'s (i.e *gRNA$^{dsx+ix+\beta Tub}$#1/Cas9*) generated with Cas9 inherited from either the mother (*pgSIT$^{♀Cas9}$*) or father (*pgSIT$^{♂Cas9}$*) were caged with 100 mature WT ♂'s for 2 days to insure that every WT ♀ was mated before WT ♂'s were removed from all cages. Then, one anesthetized mouse (similar size and weight) was put into each cage for 20 min (from 9:00 am to 9:20 am) daily for 5 days, and two times were recorded for each mosquito ♀ or ♂. First, the 'time to bite' is the time interval between a mouse placed in a cage and ♀/♂ landing on the mouse. Second, the 'feeding time' is the duration of blood feeding (*Supplementary file 1e*). When individual ♀/♂'s finished blood feeding they were removed from the cage, while unfed ♀/♂'s were kept in the cage and the blood feeding was repeated the next day. In total, each ♀/♂ had 6000 s to initiate bite and feed during 5 days. To assess whether distance affects blood feeding of *pgSIT* ♀'s, we conducted the experiment in two sizes of cages: small (24.5 × 24.5 × 24.5 cm) and large (60 × 60 × 60 cm). All experiments were repeated three times independently.

## Flight activity assays

We followed the protocol in our previous study (*Li et al., 2021*) to assess the flight activity of trans-hemizygous *pgSIT$^{♀Cas9}$* (i.e., *gRNA$^{dsx+ix+\beta Tub}$#1/Cas9*) mosquitoes (*Supplementary file 1f*).

## Mating assays

We followed a previously described protocol (*Li et al., 2021*), to assess the mating ability of both types of pgSIT males, *pgSIT$^{♀Cas9}$* and *Cas9 pgSIT$^{♂Cas9}$*, and confirm that prior matings with *pgSIT* ♂'s suppressed WT ♀ fertility (*Supplementary file 1h*).

## Fitness parameters and longevity

To assess fitness of pgSIT mosquitoes, we measured developmental times, fertility, and longevity of WT, homozygous *gRNA$^{dsx+ix+\beta Tub}$#1*, homozygous *Cas9*, and *pgSIT$^{♀Cas9}$* and *pgSIT$^{♂Cas9}$* mosquitoes. The previously published protocol was used to assess fitness parameters and longevity (*Li et al., 2021*). Briefly, all fitness parameters and longevity were assessed in three groups (three replicates) of 20 individual mosquitoes (*Supplementary file 1g*).

## Multigenerational population cage trials

To assess the competitiveness of pgSIT ♂'s, we performed discrete non-overlapping multigenerational population cage trials as described in our previous study (*Li et al., 2021*). Briefly, 3/4-day-old mature WT adult ♂'s were released along with mature (3/4 days old) pgSIT adult ♂'s at release ratios (pgSIT:WT): 1:1 (50:50), 5:1 (250:50), 10:1 (500:50), 20:1 (1000:50), and 40:1 (2000:50), with three biological replicates for each release ratio (15 cages total). One hour later, 50 mature (3/4 days old) WT adult ♀'s were released into each cage. All adults were allowed to mate for 2 days. ♀'s were then blood fed and eggs were collected. Eggs were counted and stored for 4 days to allow full embryonic development. Then, 100 eggs were randomly selected, hatched, and reared to the pupal stage, and the pupae were separated into ♂ and ♀ groups and transferred to separate cages. Three days post eclosion, ratios (pgSIT:WT) of 50 (1:1), 250 (5:1), 500 (10:1), 1000 (20:1), and 2000 (40:1) age-matched pgSIT mature ♂ adults were caged with these mature ♂'s from 100 selected eggs. One hour later, mature ♀'s from 100 selected eggs were transferred into each cage. All adults were allowed to mate for 2 days. ♀'s were blood fed, and eggs were collected. Eggs were counted and stored for 4 days to allow full embryonic development. The remaining eggs were hatched to measure hatching rates and to screen for the possible presence of transformation markers. The hatching rate was estimated by dividing the number of hatched eggs by the total number of eggs. This procedure continued for all subsequent generations.

## Mathematical modeling

To model the expected performance of pgSIT at suppressing and eliminating local *Ae. aegypti* populations, we used the MGDrivE simulation framework (*Sánchez C et al., 2020*). This framework models the egg, larval, pupal, and adult mosquito life stages with overlapping generations, larval mortality increasing with larval density, and a mating structure in which ♀'s retain the genetic material of the adult ♂ with whom they mate for the duration of their adult lifespan. We used the same egg-to-adult survival rate for WT and pgSIT eggs. The inheritance pattern of the pgSIT system was modeled within the inheritance module of MGDrivE. This module includes parameters for ♀ pupatory success (as a proxy for viability), and ♂ fertility and mating competitiveness. We simulated a randomly mixing *Ae. aegypti* population of 10,000 adults and implemented the stochastic version of MGDrivE to capture chance effects that are relevant to suppression and elimination scenarios. Density-independent mortality rates for juvenile life stages were calculated for consistency with the population growth rate in the absence of density-dependent mortality, and density-dependent mortality was applied to the larval stage following *Deredec et al., 2011*. Sensitivity analyses were conducted with 'window of protection' (mean duration for which the *Ae. aegypti* population is suppressed by at least 90%), probability of *Ae. aegypti* elimination, and reduction in cumulative potential for transmission as the outcomes, and model parameters sampled describing the construct (gRNA cutting rate and maternal deposition frequency), release scheme (number, size, and frequency of releases), ♀ viability, and ♂ fertility and mating competitiveness. Model parameters were sampled according to a latin-hypercube scheme with $2^{16}$ parameter sets and 20 repetitions per parameter set. Regression models were fitted to the sensitivity analysis data using multi-layered perceptrons as emulators of the main simulation in predicting the window of protection outcome. Subsequent simulations focused on the most sensitive parameters from the sensitivity analysis (those describing the release scheme, ♀ viability, and ♂ fertility) and were explored via factorial simulation design. Complete model and intervention parameters are listed in *Supplementary file 1l, m*.

## Transgene detection method in mosquitoes

SENSR assays were performed as described previously (*Brogan et al., 2021*) using a two-step nucleic acid detection protocol. Target sequences were first amplified in a 30-min isothermal preamplification reaction using recombinase polymerase amplification (RPA). RPA primers were designed to amplify 30 bp gRNA spacer complement regions flanked by 30 bp priming regions from the transgenic elements (*Cas9*-coding sequence or *gRNA* scaffold for *Cas9* or *gRNA*$^{dsx,ix,\beta Tub}$ construct, respectively) while simultaneously incorporating a T7 promoter sequence and 'GGG' into the 5' end of the dsDNA gene fragment to increase transcription efficiency (*Brieba et al., 2002*). The RPA primer sequences and synthetic target sequences can be found in *Supplementary file 1n*. The RPA reaction was performed at 42°C for 30 min by using TwistAmp Basic (TwistDx #TABAS03KIT). The final conditions for the optimized (28.5% sample input) RPA protocol in a 10-µl reaction are as follows: 5.9 µl rehydration buffer (TwistDx), 0.35 µl primer mix (10 µM each), 0.5 µl MgOAc (280 mM), and 50 ng of genomic DNA. The RPA reaction was then transferred to a second reaction, termed the Cas Cleavage Reaction (CCR), which contained a T7 polymerase and the CasRx ribonucleoprotein. In the second reaction, in vitro transcription was coupled with the cleavage assay and a fluorescence readout using 6-carboxyfluorescein (6-FAM). A 6-nt poly-U probe (FRU) conjugated to a 5' 6-FAM and a 3' IABlkFQ (Iowa Black Fluorescence Quencher) was designed and custom ordered from IDT (*Brogan et al., 2021*). Detection experiments were performed in 20 µl reactions by adding the following reagents to the 10 µl RPA preamplification reaction: 2.82 µl water, 0.4 µl N-2-hydroxyethylpiperazine-N'-2-ethanesulfonic acid (HEPES) pH 7.2 (1 M), 0.18 µl MgCl$_2$ (1 M), 1 µl rNTPs (25 mM each), 2 µl CasRx (55.4 ng/µl), 1 µl RNase inhibitor (40 U/µl), 0.6 µl T7 Polymerase (50 U/µl), 1 µl gRNA (10 ng/µl), and 1 µl FRU probe (2 µM). Experiments were immediately run on a LightCycler 96 (Roche #05815916001) at 37°C for 60 min with the following acquisition protocol: 5 s acquisition followed by 5 s incubation for the first 15 min, followed by 5 s acquisition and 55 s incubation for up to 45 min. Fluorescence readouts were analyzed by calculating the background-subtracted fluorescence of each timepoint and by subtracting the initial fluorescence value from the final value. Statistical significance was calculated using a one-way ANOVA followed by specified multiple comparison tests (*n* = 4). The detection speed of each gRNA was calculated from the HMF (*Brogan et al., 2021*). Raw SENSR assays data can be found in *Supplementary file 1o*.

We generated 12 gRNAs, six target regions within *Cas9* or *gRNA* transgenes, to assess the detection capabilities in an SENSR reaction for each selected gRNA using *Cas9* and *gRNA*$^{dsx,ix,\beta Tub}$ plasmids for detection of *Cas9* and *gRNA*$^{dsx,ix,\beta Tub}$ transgenic sequences, respectively. After identifying the best gRNA for detection of *Cas9* and *gRNA* sequences, we run SENSR of DNA preps made from *Ae. aegypti pgSIT*$^{\female Cas9}$ and WT mosquitoes harboring both *Cas9* and *gRNA*$^{dsx,ix,\beta Tub}$ constructs at different ratios of *pgSIT*$^{\female Cas9}$ to WT mosquitoes: 1:0; 1:1; 1:5; 1:10; 1:25; 1:50; 0:1; and NTC. We challenged the best gRNA (A4 and B3) in an SENSR assay and found that both gRNA detected transgenic DNA even at the lowest dilution and no signal was observed in both controls, only WT mosquitoes and NTC.

## Statistical analysis

Statistical analysis was performed in JMP8.0.2 by SAS Institute Inc and Prism9 for macOS by GraphPad Software, LLC. We used at least three biological replicates to generate statistical means for comparisons. p values were calculated using a two-sided Student's *t*-test with equal variance. The departure significance for survival curves was assessed with the Log-rank (Mantel–Cox) texts. Statistical comparisons of WBF for each sex were made using ANOVA on Ranks tests, followed by pairwise Wilcoxon tests with a Bonferroni correction applied for multiple comparisons. Chi-squared test was used to assess significance of departure in phonotaxis assay. All plots were constructed using Prism 9.1 for macOS by GraphPad Software and then organized for the final presentation using Adobe Illustrator.

## Ethical conduct of research

All animals were handled in accordance with the Guide for the Care and Use of Laboratory Animals as recommended by the National Institutes of Health and approved by the UCSD Institutional Animal Care and Use Committee (IACUC, Animal Use Protocol #S17187) and UCSD Biological Use Authorization (BUA #R2401).

## Reporting summary

Further information on research design is available in the Nature Research Reporting Summary linked to this article.

## Acknowledgements

We thank Judy Ishikawa for helping with mosquito husbandry and Shih-Che Weng for assistance with DNA/RNA extractions. This work was supported in part by funding from NIH (R01AI151004, R01GM132825, RO1AI148300, RO1AI175152, DP2AI152071), EPA STAR award (RD84020401), and an Open Philanthropy award (309937-0001) awarded to OSA; and the U.S. Army Research Office for the Institute for Collaborative Biotechnologies (W911NF-19-2-0026) and NIH (R01-AI165575) awards to CM; and a NIH (1R01AI143698-01A1) award to JMM. The views, opinions, and/or findings expressed are those of the authors and should not be interpreted as representing the official views or policies of the U.S. government.

## Additional information

### Competing interests

Ming Li, Nikolay P Kandul: A founder of Synvect with equity interest. The terms of this arrangement have been reviewed and approved by the University of California, San Diego in accordance with its conflict of interest policies. Omar S Akbari: A founder of Agragene, Inc with equity interest. A founder of Synvect with equity interest. The terms of this arrangement have been reviewed and approved by the University of California, San Diego in accordance with its conflict of interest policies. The other authors declare that no competing interests exist.

## Funding

| Funder | Grant reference number | Author |
|--------|------------------------|--------|
| National Institute of Allergy and Infectious Diseases | R01AI151004 | Omar S Akbari |
| Environmental Protection Agency | RD84020401 | Omar S Akbari |
| National Institute of Allergy and Infectious Diseases | 1R01AI143698-01A1 | John M Marshall |

The funders had no role in study design, data collection, and interpretation, or the decision to submit the work for publication.

## Author contributions

Ming Li, Conceptualization, Data curation, Formal analysis, Supervision, Validation, Investigation, Methodology, Writing – original draft, Writing – review and editing; Nikolay P Kandul, Data curation, Investigation, Writing – original draft; Ruichen Sun, Formal analysis, Writing – original draft, Writing – review and editing; Ting Yang, Daniel J Brogan, Formal analysis; Elena D Benetta, Igor Antoshechkin, Héctor M Sánchez C, Nicolas A DeBeaubien, YuMin M Loh, Matthew P Su, Data curation, Formal analysis; Yinpeng Zhan, Data curation, Investigation; Craig Montell, Formal analysis, Funding acquisition, Writing – review and editing; John M Marshall, Formal analysis, Funding acquisition; Omar S Akbari, Conceptualization, Supervision, Funding acquisition, Investigation, Writing – original draft, Project administration, Writing – review and editing

## Author ORCIDs

Ming Li ![ORCID] https://orcid.org/0000-0002-7578-4968
Nikolay P Kandul ![ORCID] https://orcid.org/0000-0001-7347-5558
Ting Yang ![ORCID] https://orcid.org/0000-0001-7201-4231
Elena D Benetta ![ORCID] https://orcid.org/0000-0003-2556-8500
Matthew P Su ![ORCID] https://orcid.org/0000-0002-6186-211X
Craig Montell ![ORCID] http://orcid.org/0000-0001-5637-1482
John M Marshall ![ORCID] http://orcid.org/0000-0003-0603-7341
Omar S Akbari ![ORCID] https://orcid.org/0000-0002-6853-9884

Reviewer #1 (Public Review): https://doi.org/10.7554/eLife.90199.3.sa1
Reviewer #2 (Public Review): https://doi.org/10.7554/eLife.90199.3.sa2
Reviewer #3 (Public Review): https://doi.org/10.7554/eLife.90199.3.sa3
Author Response https://doi.org/10.7554/eLife.90199.3.sa4

---

# Additional files

## Supplementary files

• Supplementary file 1. Supplementary tables. (a) gRNA and primer sequences. (b) Measurements of mosquito morphological structures. (c) Data for *dsx* and *ix* single-gene disruption. (d) Data from precision-guided sterile insect technique (pgSIT) genetic cross: *dsx*, *ix*, and *βTub* simultaneous gene disruption. (e) Data on female blood feeding assay. (f) Data on flight activity. (g) Fitness parameters and longevity. (h) pgSIT males sterilize WT females. (i) Data on population suppression. (j) RNA sequencing differential expression analysis using DeSeq2 comparing pgSIT female to WT Liverpool female. (k) RNA sequencing differential expression analysis using DeSeq2 comparing pgSIT male to WT Liverpool male. (l) Life history parameters used in mathematical modeling. (m) Network structure and training schemes for the regression surrogate models. (n) Primers and gRNA sequences for SENSR assay. (o) SENSR assay raw data.

• MDAR checklist

## Data availability

Complete sequence maps assembled in the study are deposited at https://www.addgene.org/ under ID# 200252 (gRNA$^{dsx}$, 1067B), 200251 (gRNA$^{ix}$, 1055J), and 200253 (gRNA$^{dsx,ix,βTub}$, 1067L). Illumina and Nanopore sequencing data have been deposited to the NCBI-SRA under BioProject ID PRJNA942966,

SAMN33705824–SAMN33705835; PRJNA942966, SAMN33705934. Data used to generate figures are provided in Supplementary file 1a–k. Aedes transgenic lines are available upon request.

The following dataset was generated:

| Author(s) | Year | Dataset title | Dataset URL | Database and Identifier |
|---|---|---|---|---|
| Li M, Kandul NP, Sun R, Yang T, Benetta ED, Brogan DJ, Antoshechkin I, Sánchez HM, Zhan Y, DeBeaubien NA, Loh YM, Su MP, Montell C, Marshall JM, Akbari OS | 2023 | Targeting sex determination to suppress mosquito populations | https://www.ncbi.nlm.nih.gov/bioproject/?term=PRJNA942966 | NCBI BioProject, PRJNA942966 |

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
