## [Editor Report · eLife assessment]

This **valuable** paper builds on a method, previously conceptualized and validated, of genetic control for insect populations. The method, called pgSIT, uses integrated CRISPR-Cas9 based constructs to generate, in certain combinations of genotypes, mutations that cause both male sterility and female inviability. Release of such genotypes in sufficiently large numbers can lead to an inundation of a local insect population with sterile males and this can lead to localised population suppression, which represents an effective method of control for problematic insect populations. The data are **convincing** and will be of interest to anyone working on vector control strategies.

---

## [Referee Report · Reviewer #1 (Public Review)]

Precision guided sterile insect technology (pgSIT) is a means of mosquito vector control that aims to simultaneously kill females while generating sterile males for field release. These sterile males are expected to mate with 'wild' females resulting in very few eggs being laid or low hatching rates. Repeated releases are expected to result in the suppression of the mosquito population. This method avoids cumbersome sex-sorting while generating the sterile males. Importantly, until release, the two genetic elements that bring about female lethality and male sterility - the Cas9 and the gRNA carrying mosquitoes - are maintained as separate lines. They are crossed only prior to release, and therefore, this approach is considered to be more safe than gene drives.

The authors had made a version of this pgSIT in their 2021 paper where they targeted *β-Tubulin 85D*, which is only expressed in the male testes and its loss-of-function results in male sterility. In that pgSIT, they did not have female lethality, but generated flightless females by simultaneously targeted *myosin heavy chain,* which is expressed only in the female wings. Here the authors argue, that the survival of females is not ideal, and so modify their 2021 approach to achieve female lethality/sterility.

To do this, they target two genes - the female specific isoform of Dsx and intersex. They use multiple gRNAs against these genes and validate their ability to cause female lethality/sterility. Having verified that these do indeed affect female fertility, they combine gRNAs against Dsx and ix to generate female lethality/sterility and use *β-Tubulin 85D* to generate male sterility (previously validated). When these gRNA mosquitoes are crossed to Cas9 and the progeny crossed to WT (the set-up for pgSIT), they find that very few eggs are laid, larval death is high, and what emerges are males or intersex progeny that are sterile.

As this is the requirement for pgSIT, the authors then test if it is able to induce population suppression. To do this, they conduct cage trials and find that only when they use 20:1 or 40:1 ratio of pgSIT:WT cages, does the population crash in 4-5 generations. They model this pgSIT's ability to suppress a population in the wild. Unfortunately, I was not able to assess what parameters from their pgSIT were used in the model and therefore the predicted efficacy of their pgSIT, (though the range of 0-.1 is not great, given that the assessment is between 0-0.15).

Finally, they also develop a SENSR with a rapid fluorescence read-out for detecting the transgene in the field. They show that this sensor is specific and sensitive, detecting low copy numbers of the transgene. This would be important for monitoring any release.

Overall, the data are clear and well presented.

Comments on revised version:

The authors have addressed the major issues raised by reviewers related to off target effects, writing and figures, and comparisons with other vector control methods and claims made in passing.

---

## [Referee Report · Reviewer #2 (Public Review)]

This is a thorough and convincing body of work that represents an incremental but significant improvement on iterations of this method of CRISPR-based Sterile Insect Technique ('pgSIT'). In this version, compared to previous, the authors target more genes than previously, in order to induce both female inviability (targeting the genes intersex and doublesex, compared to fem-myo previously) and male sterility targeting a beta-tubulin, as previously in the release generation.

The characterization of the lines is extensive and this data will be useful to the field. However, what is lacking is some context as to how this formulation compares to the previous iteration. Mention is made of the possible advantage of removing most females, compared to just making them flightless (as previously) but there is no direct comparison, either experimental, or theoretical i.e. imputing the life history traits into a model. For me this is a weakness, yet easily addressed. In a similar vein, much is made in alluding to the 'safety concerns of gene drive' and how this is a more palatable half-way house, just because it has CRISPR component within it; it is not. It would be much more sensible, and more informative, to compare this pgSIT technology to RIDL (release of insects carrying a dominant lethal), which is essentially a transgene-based version of the Sterile Insect Technique, as is the work presented here.

The authors achieve impressive results and show that these strains, under a scenario of high levels of release ratios compared to WT, could achieve significant local suppression of mosquito populations. The sensitivity analysis that examines the effect of changing different biological or release parameters is well performed and very informative.

The authors are honest in acknowledging that there are still challenges in bringing this to field release, namely in developing sexing strains and optimizing release strategies.

---

## [Referee Report · Reviewer #3 (Public Review)]

The manuscript by Li et al. presents an elegant application of sterile insect technology (pgSIT) utilizing a CRISPR-Cas9 system to suppress mosquito vector populations. The pgSIT technique outlined in this paper employs a binary system where Cas9 and gRNA are conjoined in experimental crosses to yield sterile male mosquitoes. Employing a multiplexed strategy, the authors combine multiple gRNA to concurrently target various genes within a single locus. This approach successfully showcases the disruption of three distinct genes at different genomic positions, resulting in the creation of highly effective sterile mosquitoes for population control. The pioneering work of the Akbari lab has been instrumental in developing this technology, previously demonstrating its efficacy in *Drosophila* and *Aedes aegypti*.

By targeting the female-specific splice isoform (exon-5) of doublesex in conjunction with intersex and β-tubulin, the researchers induce female lethality, leading to a predominance of sterile male mosquitoes. This innovation is particularly noteworthy as the deployment of sterile mosquitoes on a large scale typically requires substantial investment in sex sorting. However, this study circumvents this challenge through genetic manipulation.

---

## [Author Response]

The following is the authors’ response to the original reviews.

**eLife assessment**

This important paper builds on a method, previously conceptualized and validated, of genetic control for insect populations. The method, called pgSIT, uses integrated CRISPR-Cas9 based constructs to generate, in certain combinations of genotypes, mutations that cause both male sterility and female inviability. Release of such genotypes in sufficiently large numbers can lead to an inundation of a local insect population with sterile males and this can lead to localised population suppression, which represents an important method of control for problematic insect populations. The data are convincing and will be valuable to anyone working on vector control strategies.

**Public Reviews:**

**Reviewer #1 (Public Review):**
Precision guided sterile insect technology (pgSIT) is a means of mosquito vector control that aims to simultaneously kill females while generating sterile males for field release. These sterile males are expected to mate with 'wild' females resulting in very few eggs being laid or low hatching rates. Repeated releases are expected to result in the suppression of the mosquito population. This method avoids cumbersome sex-sorting while generating the sterile males. Importantly, until release, the two genetic elements that bring about female lethality and male sterility - the Cas9 and the gRNA carrying mosquitoes - are maintained as separate lines. They are crossed only prior to release, and therefore, this approach is considered to be more safe than gene drives.The authors had made a version of this pgSIT in their 2021 paper where they targeted *β-Tubulin 85D*, which is only expressed in the male testes and its loss-of-function results in male sterility. In that pgSIT, they did not have female lethality, but generated flightless females by simultaneously targeted *myosin heavy chain,* which is expressed only in the female wings. Here the authors argue, that the survival of females is not ideal, and so modify their 2021 approach to achieve female lethality/sterility.To do this, they target two genes - the female specific isoform of Dsx and intersex. They use multiple gRNAs against these genes and validate their ability to cause female lethality/sterility. Having verified that these do indeed affect female fertility, they combine gRNAs against Dsx and ix to generate female lethality/sterility and use *β-Tubulin 85D* to generate male sterility (previously validated). When these gRNA mosquitoes are crossed to Cas9 and the progeny crossed to WT (the set-up for pgSIT), they find that very few eggs are laid, larval death is high, and what emerges are males or intersex progeny that are sterile.As this is the requirement for pgSIT, the authors then test if it is able to induce population suppression. To do this, they conduct cage trials and find that only when they use 20:1 or 40:1 ratio of pgSIT:WT cages, does the population crash in 4-5 generations. They model this pgSIT's ability to suppress a population in the wild. Unfortunately, I was not able to assess what parameters from their pgSIT were used in the model and therefore the predicted efficacy of their pgSIT, (though the range of 0-.1 is not great, given that the assessment is between 0-0.15).

We express our sincere appreciation for the valuable comments received. A wide range of ♀ viability and ♂ fertility values were explored in the model. The results determined that: “Achieving a ≥90% probability of elimination places slightly tighter restrictions on ♀ viability and ♂ fertility - a safe ballpark being ♀ viability and ♂ fertility both in the range 0-0.10, given a release scheme of ~26 releases of 250 pgSIT eggs per wild adult (Fig. 4B). These results suggest a target product profile for pgSIT to be ♀ viability and ♂ fertility both in the range 0-0.10.” A subsequent sentence has been added pointing out how the described pgSIT strain falls within this range: “The pgSIT strain described here falls well within these bounds, with ♀ viability of 0 and ♂ fertility of ~0.01.” The parameters of the described pgSIT strain are also listed throughout the paper and quoted here: “Cas9 in combination with gRNAdsx,ix,βTub induces either the lethality or transformation of pgSIT ♀’s into sterile unfit ⚥’s.” And: “Firstly, we determined that pgSIT males were not 100% sterile, with an estimated ~1% still producing some progeny.”

Finally, they also develop a SENSR with a rapid fluorescence read-out for detecting the transgene in the field. They show that this sensor is specific and sensitive, detecting low copy numbers of the transgene. This would be important for monitoring any release.Overall, the data are clear and well presented. The manuscript is well written (albeit likely dense for the uninitiated!). I had concerns about the efficacy of generating the pgSIT animals - the overall number of eggs hatched from the gRNA (X) Cas9 cross appears to be low, therefore, very large numbers of parental animals would have to be reared and crossed to obtain enough sterile males for the SIT. In addition to this, I was concerned about the intersex progeny that can blood-feed. These could potentially contribute to the population and it would be useful to see the data that suggest that these numbers are low and the animals will not be competent in the field.
**Reviewer #2 (Public Review):**
This is a thorough and convincing body of work that represents an incremental but significant improvement on iterations of this method of CRISPR-based Sterile Insect Technique ('pgSIT'). In this version, compared to previous, the authors target more genes than previously, in order to induce both female inviability (targeting the genes intersex and doublesex, compared to fem-myo previously) and male sterility targeting a beta-tubulin, as previously in the release generation.The characterization of the lines is extensive and this data will be useful to the field. However, what is lacking is some context as to how this formulation compares to the previous iteration. Mention is made of the possible advantage of removing most females, compared to just making them flightless (as previously) but there is no direct comparison, either experimental, or theoretical i.e. imputing the life history traits into a model. For me this is a weakness, yet easily addressed. In a similar vein, much is made in alluding to the 'safety concerns of gene drive' and how this is a more palatable half-way house, just because it has CRISPR component within it; it is not. It would be much more sensible, and more informative, to compare this pgSIT technology to RIDL (release of insects carrying a dominant lethal), which is essentially a transgene-based version of the Sterile Insect Technique, as is the work presented here.

We express our sincere appreciation for the valuable comments received. A wide range of ♀ viability and ♂ fertility values were explored in the model. Given the intricate nature of this study and taking into account the recommendations provided by multiple reviewers and the editor, we have eliminated superfluous comparisons among various methodologies.

The authors achieve impressive results and show that these strains, under a scenario of high levels of release ratios compared to WT, could achieve significant local suppression of mosquito populations. The sensitivity analysis that examines the effect of changing different biological or release parameters is well performed and very informative.The authors are honest in acknowledging that there are still challenges in bringing this to field release, namely in developing sexing strains and optimizing release strategies - a question I have here is how to actually release eggs, and could variability in the efficiency of this aspect be modelled in the sensitivity analysis? It seems to me like this could be a challenge and inherently very variable.

We really appreciate comments. Several approaches are available to release eggs - either in pre-existing breeding sites in the field, or in artificial breeding sites (e.g., cups). We have added a sentence in the Discussion section to highlight that this is an area requiring further research: “Secondly, studies are required to determine the survival and mating competitiveness of released pgSIT males under field conditions, and to optimize their release protocol.” Regarding the efficiency of egg releases, the following sentence in the modeling results section has been added: “We assume released eggs have the same survival probability as wild-laid eggs; however if released eggs do have higher mortality, this would be equivalent to considering a smaller release.” As stated in the modeling results (and depicted in Figure 4 and Supplementary Figure 5): “Suppression outcomes were found to be most sensitive to release schedule parameters (number, size and interval of releases), ♂ fertility and ♀ viability.” It follows that suppression outcomes are equivalently sensitive to the efficiency of an egg release.

**Reviewer #3 (Public Review):**
Summary and Strengths:The manuscript by Li et al. presents an elegant application of sterile insect technology (pgSIT) utilizing a CRISPR-Cas9 system to suppress mosquito vector populations. The pgSIT technique outlined in this paper employs a binary system where Cas9 and gRNA are conjoined in experimental crosses to yield sterile male mosquitoes. Employing a multiplexed strategy, the authors combine multiple gRNA to concurrently target various genes within a single locus. This approach successfully showcases the disruption of three distinct genes at different genomic positions, resulting in the creation of highly effective sterile mosquitoes for population control. The pioneering work of the Akbari lab has been instrumental in developing this technology, previously demonstrating its efficacy in *Drosophila* and *Aedes aegypti*. By targeting the female-specific splice isoform (exon-5) of doublesex in conjunction with intersex and β-tubulin, the researchers induce female lethality, leading to a predominance of sterile male mosquitoes. This innovation is particularly noteworthy as the deployment of sterile mosquitoes on a large scale typically requires substantial investment in sex sorting. However, this study circumvents this challenge through genetic manipulation.Weaknesses:One notable concern arising from this manuscript pertains to the absence of data regarding the potential off-target effects of the gRNA. Given the utilization of multiple gRNA, the risk of unintended mutations in non-target areas of the genome increases. With around 1% of males still capable of producing fertile offspring, understanding the frequency of unintended genome targeting becomes crucial. Such mutations could potentially become fixed within the natural population.

We express our sincere appreciation for the valuable comments received and fully agree with the reviewer regarding the importance of understanding the frequency of unintended genome targeting. However, the likelihood of off-target effects becoming fixed within the population is exceedingly low. To mitigate potential negative impacts, we employed CHOPCHOP V3.0.0 (https://chopchop.cbu.uib.no) for the selection of gRNAs, which will specifically tminimize the occurrence of genomic off-target cleavage events. Furthermore, our releasing process will be carried out in multiple rounds. In the event that an undesired mutant is introduced into the local population, the mutated gene will either be quickly eradicated through subsequent rounds of releases or be naturally eliminated through the process of natural selection over time.

The experiments are well-conceived, featuring suitable controls and repeated trials to yield statistically significant data. However, a primary issue with the manuscript lies in its data presentation. The authors' graphical representations are intricate and demand considerable attention to discern the nuances, especially due to the striking similarity between the symbols representing different genotypes. As it stands, the manuscript primarily caters to experts within the field, thereby warranting improvements in data visualization for broader comprehension.

We appreciate the comment. However, as this work is indeed complex and intricate and as there is limitations imposed by the publisher on data visualizations (i.e. number of figures in the main text, etc.) we have tried our best for presenting our data in full.

All three reviewers were appreciative of the work presented in this manuscript. There were some common concerns that we shared, that the authors could consider revising. They are listed below.Essential revisions:1. Formal comparison with the previous/other methods: The authors make many statements that compare this pgSIT with their previous method, gene drives, or with RIDL. We suggest that they focus their comparisons within the scope of data and avoid comparisons between RIDL, gene drive, and pgSIT that are based on perceptions of these methods. It would be useful if, for example, they could impute life history traits and demonstrate this pgSIT's efficacy over their previous versions.

We express our sincere appreciation for the valuable comments received. We have removed the unnecessary comparisons between different methods, please review the revised version.

2. Writing and presentation of figures: The authors should please take advantage of the eLife format and unpack each sentence/figure so that it's accessible to readers outside this field.

We appreciate your comment, and we have implemented some necessary changes based on your suggestions.

3. Data to support claims made in passing: There are many instances, such as detailed in the reviews (and the entire second paragraph in the discussion) that are not supported by data. The authors should either provide that data or not make these claims.

Thank you for the comment. We have removed these claims.

4. Off target effects: There is the formal possibility that off target effects that might get fixed in the population. Could the authors please address this in the discussion.

We appreciate the comment and fully agree with the reviewer regarding the importance of understanding the frequency of unintended genome targeting. However, the likelihood of off-target effects becoming fixed within the population is exceedingly low. We have address this in the discussion.

“Even though mutations could potentially become fixed within the natural population, the likelihood of off-target effects becoming fixed within the population is exceedingly low. To mitigate potential negative impacts, we employed CHOPCHOP V3.0.0 (https://chopchop.cbu.uib.no) for the selection of gRNAs, specifically to minimize the occurrence of genomic off-target cleavage events. Furthermore, our releasing process will be carried out in multiple rounds. Even in the event that an undesired mutant is introduced into the local population, it will either be completely eradicated through subsequent rounds of releases or be naturally eliminated through the process of natural selection over time.”

Aside from this, we ask that the authors please pay attention to the detailed reviews.
**Reviewer #1 (Recommendations For The Authors):**
The writing: Each sentence is packed with information and while this is fine for those immersed in the field, it might be dense for those who are not. There are a lot of nuances in such an approach and clearly laying it out for the reader is important. The authors should unpack some of these sentences to make their work more accessible.

Thank you for the comment. We have unpacked some of sentences, please review the revised version.

It will help to have a schematic linked to the introduction about how these mosquitoes are designed to be used. Which strains would be scaled up in the lab, which ones (and what stage) could be released, and in which animal/generation they expect sterility or lethality. This would be useful while interpreting the schematics of the genetic crosses in the rest of the figures (1B, 2B). Li et al 2021 has something to this effect. I say this particularly because in the text, 'pgSIT' is used to refer to both the lab stocks and the F1s.

We really appreciate the suggestion to incorporate a schematic into the introduction to clarify the intended use of these mosquitoes. Taking into account all the suggestions, we would like to keep textual descriptions and context provided within the manuscript, which, together with Figures 1B and 2B, illustrate our intentions. Nevertheless, we value your input and have taken other feedback into account to improve the overall quality of the content.

Because Figure 1A depicts all the gRNAs I thought that's what they were testing in the first results section. But the legends seems to suggest that the individual gRNAs have been tested. Such issues will be sorted with attention to the writing. It would also be nice to have Figure 2A here.

We apologize for any misunderstanding. Figure 1A displays two gRNA constructs: one for dsx (comprising 4 gRNAs) and another for ix (with 2 gRNAs). All of these gRNAs were tested in the initial results section. Subsequently, we engineered the final gRNA construct, denoted as gRNAdsx,ix,βTub, which combines the effective gRNAs described earlier (3 targeting dsx and 1 targeting ix, as illustrated in Supplementary Figure 2).

It wasn't clear to me how egg laying percentages were calculated or what it means.

We appreciate your comment. Female fecundity depends on the egg output (egg laying percentage) and the egg hatching rate, since insect female can lay unfertalized eggs that does not hatch. Egg laying percentages were calculated by dividing the numbers of laid eggs by a test female group by that of the control female group that laid the highest egg number. This procedure is called normalization and enable relative comparison of laid egg number.

How is hatching at times more than laying?

When a female group laid a small egg number but the high percentage of those eggs hatched.

Calling something 'intersex': The authors are assessing intersex by malformed genitalia, maxillary palps and ovaries. But the genitalia defects in Fig1D were not clear to me. Can the authors show better images? While the MP snd ovary phenotypes were clear, it would be nice to see these quantified - what proportion of the females have each/some/all of these phenotypes? It would be nice to see this quantified. (They have some of this in the supplementary table).

We express our gratitude for the comment received and acknowledge the issue regarding the clarity of the images. It is important to note that these photographs represent the highest level of clarity achieved thus far. We value your interest in the quantification of the observed phenotypes. However, due to certain constraints, we were unable to quantify the proportions for all the females, and we did not retain all the samples needed for this specific quantification.

It's interesting that 50% of the intersex don't blood-feed - is this because they do not have appropriately formed stylets? It would be important to quantify the number of hatch-able eggs. This is particularly important in the context of field application and should ideally be included in the mathematical modelling. In the discussion, the authors mention that they are not able to host-seek and a variety of other behaviours - these data should be presented as it would be important for assessing the efficacy of the pgSIT.

Thank you for the comment. We did not find the mutant stylets from these intersex mosquitoes. We agree with the reviewer that the number of hatchable eggs is particularly important in the context of field application. Indeed, the number of hatchable eggs is what was considered in the mathematical modeling. We did a blood feed assay (small cage and big cage) for host seeking behavior. Data were presented in Supplementary Table 5.

At the end of the first results section, the authors state, "Taken together, these findings reveal that ♀-specific lethality and/or ⚥..." But I don't see data that show female-specific lethality until Figure 2C.

Thank you for pointing out this. In order to describe our results clearly, we have deleted “♀-specific lethality and/or”

In the combined gRNA mosquito (the pgSIT), they find that the cross between the gRNA and Cas9 results in very few eggs being laid, high larval death, and what emerges are males. This suggests that it would be a poor pgSIT, right? You'd have to set up huge crosses to get enough males emerging in the wild to mate with WT females to bring about population suppression. Could the authors comment on this?

We appreciate the comment. Even in the presence of imperfections, such as reduced egg production resulting from the gRNA and Cas9 cross and the necessity of extensive mating to obtain an adequate number of males, population suppression is very promising with the pgSIT, both in terms of the potential to eliminate a mosquito population, or to suppress it to an extent that would largely interrupt disease transmission. It's worth noting that our current efforts serve as a validation of the system before its potential large-scale application, because we have demonstrated that removing females by disrupting sex determinate genes is possible with pgSIT, which can inform the development of such systems in other species in the future.

If I'm reading Figure 2C right, the authors have combined the results from two types of crosses in the last two plots: (1) the Cas9 (X) gRNA mosquitoes and (2) the progeny from these crossed to WTs. This is not ideal. I would suggest the authors unpack the text around this data and plot it separately.

We really appreciate the comment here, the panel 2C depicts the phenotypic data of the F1 progeny generated by the cross of the parents indicated below the X axis: egg-to-adult survival, larval death, sex ratios, and fertility. The fertility of F1 progeny is the major phenotypic feature for the project. To assess the fertility of the surviving F1 progeny, we had to cross the F1 females and males to WT males and females, respectively and assess the hatching rate of produced eggs before sacrificing emerged larvae and unhatched eggs. It's important to note that mosquito females can lay unfertilized eggs that fail to hatch.

The text around 2F needs to be more explanatory. There are lots of labels in the figure that are not referred to, making it difficult to follow the data.

We have gone through and expanded many of the figure legends and modified some figures to help make them more understandable.

The supplementary figure numbering is off.

We really appreciate the comment. The supplementary figure numbering have been fixed.

I cannot comment on Figure 4 as this is outside my expertise. However, I do feel that some attention to the writing might help make the approach more accessible to the invested advanced lay-person.

We appreciate the comment, and we re-wrote some of the sentences describing Figure 4.

**Reviewer #2 (Recommendations For The Authors):**
Line 49 'resistances' is a strange plural.

Corrected. Thank you so much!

the genitive, used with the sex symbols throughout, looks very weird e.eg line 60, 66 etc. Also the intersex symbol, on my copy at least, just prints as a square

These have been fixed in the revised version. Thank you so much!

Line 74 syntax (...: the spread of...") seems off

Corrected. Thank you for pointing out this.

Line 80-81 " to address some of the challenges with gene drives, pgSIT also leverages....." this is a straw man/red herring argument, and simply does not follow. It is this element that I raised above in the public review. See also line 84 'gene drive safety concerns'.

Thank you, we have re-wrote the paragraph.

Line 128 "the induced phenotypes were especially strong in intersex individuals" - this is a curious statement since, if intersex, they are by definition already showing a strongly induced phenotype

We apologize for the lack of clarity and have updated the text, we have deleted “the induced phenotypes were especially strong in intersex individuals”, to be more explicit, now stating “These gRNAdsx/+; Cas9/+ ⚥ exhibited multiple malformed morphological features, such as mutant maxillary palps, abnormal genitalia, and malformed ovaries”

The extent and completeness of the supplementary data is appreciated but there needs to be some statistical tests applied to back up statements like 'showed normal fertility' (line 138) or wind lengths 'were a bit larger'. None seem to have been applied.

We appreciate the comment. We've removed these sentences in the new version.

Supp Fig 4 - on left of panel C there is a small blue square at dsx locus that is unexplained. What is this?

Thank you for pointing this. It was a mistake, we have removed the small blue square from Sup Fig4.

Line 182 the reduction in flight activity in release genotype of pgSIT males - is it only those coming with the maternal source of Cas9 that are plotted (only pink dots)?

We appreciate the comment. pgSIT males, regardless of whether they originate from a maternal or paternal source of Cas9, exhibit a similar reduction in flight activity compared to wild-type (WT) males.

Figure 3A legend - I think there is a typo that says males were fed

Corrected. Thank you for pointing this out.

“♂’s” to “♀’s”

On the window of protection (WOP) plots (e.g. supp fig 12) what is the unit on Y-axis for WOP? It goes from 0-1, as if it were probability, but I was expecting some duration.

Thanks for the comment. The y-axis for WOP in Supp Fig 12 had been normalized unnecessarily. It has now been corrected to span from 0 to 5 years.

Fig 4B blue (line) on blue(shading) is impossible to decipher on my copy

Thank you for pointing this out. We have changed the colors of the traces (population dynamics), made the window of protection line thicker, and have made the shading less opaque to make the population dynamics in this figure clearer.

Line 250 and 252: supp Fig 13 (not 12)

Corrected. Thank you for pointing this out.

Line 279 "potentially a more widespread effect of sex determination genes than previously expected" - I simply don't see how this is so, or why there is the need to make such a claim. Dsx is known to underpin almost of somatic determination of sex-specific morphologies, in a range of insects.

We appreciate the comment. We have delete the sentence:

“Taken together, these observations indicate a potentially more widespread effect of sex determination genes than previously expected, though regardless.”

Line 320 "We would expect pgSIT to be regulated similarly to Oxitec's RIDL" because they are similar, which goes to my main point above about more appropriate context, and this warrants some direct attention to a comparison of the efficacy.

We appreciate the comment. We have delete these sentences:

“We would expect pgSIT to be regulated similarly to Oxitec's RIDL technology (Spinner et al., 2022), which has already been successfully deployed in numerous locations, including the United States.”

Was there a minimal performance advantage with strain #1 with the triple locus g-RNA suite, over the other two strains? Am just curious as to why one was chosen over the other

We appreciate the comment. There was no performance advantage with the strain #1 over the other two strains.